# PERK: LONG-CONTEXT REASONING AS PARAMETER-EFFICIENT TEST-TIME LEARNING

**Zeming Chen**    **Angelika Romanou**    **Gail Weiss**    **Antoine Bosselut**

EPFL

email {zeming.chen, antoine.bosselut}@epfl.ch

## ABSTRACT

Long-context reasoning requires accurately identifying relevant information in extensive, noisy input contexts. In this work, we propose PERK (**P**arameter **E**fficient **R**easoning over **K**nowledge), a scalable approach for learning to encode long contexts using gradient updates at test time. Specifically, PERK employs two nested optimization loops in a meta-training phase. The inner loop rapidly encodes contexts into a low-rank adapter (LoRA) that serves as a parameter-efficient memory module for the base model. Concurrently, the outer loop learns to use the updated adapter to accurately recall and reason over relevant information from the encoded long context. Our evaluations on several long-context reasoning tasks show that PERK significantly outperforms the standard long-context finetuning, achieving average absolute performance gains of up to 20% for Qwen-2.5 (0.5B & 7B) on synthetic and real-world long-context reasoning. PERK also maintains its advantages across model scales and families. Compared to specialized long-context LLMs, PERK matches or surpasses their performance. Finally, our analyses show PERK is more robust to reasoning complexity, length extrapolation, and the positions of relevant information in contexts. https://perk-long-context.web.app

## 1 INTRODUCTION

Large language models (LLMs) struggle with identifying and reasoning over relevant information in long, potentially noisy, input contexts (*i.e.*, long-context reasoning; Anil et al., 2022; Lee et al., 2024; Li et al., 2024; Liu et al., 2023; Xu et al., 2024), including modern models with substantial context windows (Kuratov et al., 2024; Levy et al., 2024; Hsieh et al., 2024). These challenges arise because longer contexts contain more irrelevant *distractor* information, and may also be attached to more complex reasoning problems (*i.e.*, multiple hops), both of which degrade LLM performance (Chen et al., 2023b). Furthermore, these challenges are exacerbated by positional biases in long-context models, which focus attention on the beginning or end of contexts, neglecting intermediate information (Liu et al., 2024).

In this work, we address these challenges of long-context reasoning by reframing it as test-time learning, enabling models to encode context at inference time using gradient-based parameter adaptation. Our approach, Parameter-Efficient Reasoning over Knowledge (PERK), processes and compresses lengthy sequences as a *batch* (or multiple batches) of shorter segments of the long context, rather than processing the full sequence token-by-token. The updated model can then use the newly stored parametric information (and its initial parametric knowledge) to answer relevant questions. The original context is then discarded in this answering step as it is now encoded parametrically by the model. To train PERK, we operate two nested optimization loops: the inner loop, which is also used at test time, encodes the long context segment batches into model parameters, while the outer loop learns to respond to queries using the updated parameters representing the context.

However, bi-level optimization algorithms for training test-time learning methods entail prohibitive memory overhead (Finn et al., 2017; Chen et al., 2023b; Hu et al., 2023), particularly when updating all model parameters, because they require backpropagating through a complete multi-step optimization trajectory. To reduce memory overhead, PERK (1) internalizes the context into a Low-Rank Adapter

(LoRA; Hu et al., 2022) rather than the initial model parameters, and (2) backpropagates the outer loop over only a truncated trajectory from the final few inner-loop adaptation steps (Shaban et al., 2019). Consequently, PERK scales efficiently at training, supporting larger models and longer contexts.

We compare PERK to models finetuned on long contexts for classical in-context reasoning (FT-ICR) on multiple *Needle-in-a-Haystack* benchmarks, demonstrating PERK's advantages. For Qwen-2.5-0.5B with a native 32K context window, PERK surpasses FT-ICR baselines by a 20% absolute improvement. PERK also outperforms specialized long-context LLMs (Gao et al., 2025a; Yang et al., 2025) with over 7B parameters that have undergone extensive long-context training, achieving the highest accuracy when generalizing to contexts up to 128k tokens. As relevant facts in *Needle-in-a-Haystack* problems are often easily distinguishable from distractor text, we also propose more challenging evaluation settings where relevant facts are surrounded by distributionally-similar distractor facts (*e.g.*, *Drops-in-the-Ocean*, HotPotQA, TriviaQA). In these settings, PERK models again outperform the FT-ICR baselines, and even surpass commercial LLMs (*e.g.*, GPT, Gemini) on benchmarks for long-context symbolic reasoning. PERK also works effectively across many model scales (0.1B, 0.5B, 8B) and families (GPT-2, Qwen-2.5, LLaMA-3.1/3.2).

Finally, our analysis also shows PERK's advantages yield strong test-time length extrapolation. PERK trained on 8K-token contexts maintains decent performance on 64K and 128K windows, consistently surpassing FT-ICR length generalization, and even outperforming models with native 512K and 1M context windows. Finally, because PERK encodes sequences as a permutation-invariant *batch* of context window segments (rather than maintaining the full contiguous context), it generalizes across contexts where relevant information is variably distributed along context positions, unlike FT-ICR's dramatic drops in performance (up to 90%) when relevant information shifts positions at test time.

## 2 BACKGROUND AND PRELIMINARIES

**Causal Language Models.** *Causal language models* (CLM) $f_{\boldsymbol{\theta}}$ (Radford & Narasimhan, 2018) with parameters $\boldsymbol{\theta}$ are typically trained with an autoregressive learning objective such as the *token-level negative log-likelihood* (NLL). For a dataset $\mathcal{D} = \{x\}$ with $N_{\mathcal{D}}$ sequences, this loss is defined as:

$$\mathcal{L}_{\text{NLL}}(\boldsymbol{\theta}, \mathcal{D}) = -\frac{1}{N_{\mathcal{D}}} \sum_{x \in \mathcal{D}} \sum_{t=1}^{\mathcal{T}} \log p_{\boldsymbol{\theta}}(x_t \mid \boldsymbol{x}_{<t}). \tag{1}$$

The initial parameters $\boldsymbol{\theta}$ are optimized iteratively via an optimization algorithm $\mathcal{A}lg : (\mathcal{L}, \mathcal{D}, \boldsymbol{\theta}, \boldsymbol{h}) \mapsto \boldsymbol{\theta}'$, where $\boldsymbol{h}$ are hyperparameters, such as optimizer choice and training length.

**Reasoning with CLMs.** We denote *reasoning* problems as tuples $r = (\mathcal{K}, \boldsymbol{q}, y)$ of context $\mathcal{K}$, question $\boldsymbol{q}$, and target response $y$ (*e.g.*, a question-answering dataset like HotpotQA; Yang et al., 2018). With the pretrained CLMs's ability to reason over facts in input contexts with reasonable success (Clark et al., 2020), prompt-based *in-context reasoning* (ICR) has become popular for solving reasoning tasks. In this setting, $\mathcal{K}$ and $\boldsymbol{q}$ are combined into a single prompt $\boldsymbol{x}(r)$. The model's parameters are optimized to predict the tokens of the correct answer:

$$\mathcal{L}_{\text{reason}}(\boldsymbol{\theta}, \mathcal{D}) = -\frac{1}{N_{\mathcal{D}_y}} \sum_{r=(\mathcal{K}, \boldsymbol{q}, y) \in \mathcal{D}} \sum_{t=1}^{\mathcal{T}_y} \log p_{\boldsymbol{\theta}}(y_t \mid \boldsymbol{x}(r), y_{<t}) \tag{2}$$

where $\mathcal{D}$ is the training corpus, $\mathcal{T}_y$ is the length of an example answer, and $N_{\mathcal{D}_y}$ is the total length of all answers in the dataset.

**Reasoning as Test-time Learning.** In this setting, models simulate reasoning by first encoding the given context $\mathcal{K}$ into a CLM's parameters $\boldsymbol{\theta}$, and then using the updated parameters $\phi(\boldsymbol{\theta}, \mathcal{K})$ to generate a response to the given question $\boldsymbol{q}$ (Chen et al., 2023b). The context is encoded by adapting the model's parameters using a CLM objective over the context: $\phi(\boldsymbol{\theta}, \mathcal{K}) = \mathcal{A}lg(\mathcal{L}_{\text{NLL}}, \mathcal{K}, \boldsymbol{\theta}, \boldsymbol{h})$. We then use $f_{\phi}$ to generate answers for any relevant questions $\boldsymbol{q}$.

Test-time learning methods for reasoning stem from a key observation: CLMs can store and retrieve diverse information in their parameters from pre-training (Roberts et al., 2020; Bayazit et al., 2024). Yet these same models struggle with long-context problems (Hsieh et al., 2024; Kuratov et al., 2024; Xu et al., 2024; Levy et al., 2024), despite the contexts containing far less information than what is seen during pre-training. CLMs may reason better with knowledge in their parameters than in context, motivating methods that encode new context parametrically for reasoning.

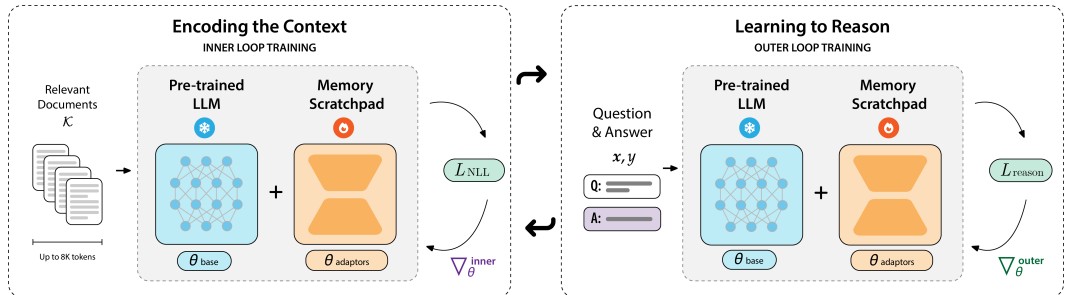

Figure 1: **Meta-learning `PERK` for long-context reasoning.** The training procedure involves a nested inner and outer loop. The inner loop optimizes the likelihood of a batch of long context segments with respect to the parameters of the LoRA-based memory scratchpad. In the outer loop, the model uses the encoded information in the memory scratchpad to answer questions. In both cases, only the memory scratchpad parameters are updated while the base LLM's parameters are frozen.

**Training a CLM for Test-time Learning.** Given a distribution $R$ of possible reasoning problems, the optimal parameters $\boldsymbol{\theta}^*$ for a test-time learning CLM minimize the expected reasoning loss of the CLM's adaptation to each reasoning problem $r \sim R$:

$$\boldsymbol{\theta}^* = \arg\min_{\boldsymbol{\theta}} \ \mathbb{E}_{r \sim R} \left[ \mathcal{L}_{\text{reason}}\Big( \phi(\boldsymbol{\theta}, \mathcal{K}), \{(\boldsymbol{q}, y)\} \Big) \right]$$

where $\phi(\boldsymbol{\theta}, \mathcal{K})$ is a CLM adaptation of $\boldsymbol{\theta}$ to the context $\mathcal{K}$: $\phi(\boldsymbol{\theta}, \mathcal{K}) = \mathcal{A}lg(\mathcal{L}_{\text{NLL}}, \mathcal{K}, \boldsymbol{\theta}, \boldsymbol{h})$.

Meta-learning algorithms, such as MAML (Finn et al., 2017), can be used for training effective test-time learning methods. In MAML, we obtain a meta-model $f_{\boldsymbol{\theta}}$ via bi-level optimization: the *outer loop* samples batches of reasoning problems from a training dataset of reasoning problems, the *inner loop* obtains the adaptations of $\boldsymbol{\theta}$ to each problem's context $\mathcal{K}$, and the *outer loop* then optimizes the meta-parameters based on the performance of each $f_{\phi(\boldsymbol{\theta}, \mathcal{K})}$ on the corresponding questions. The outer loop optimization backpropagates through the entire inner loop, an operation that becomes increasingly expensive with more parameters and inner loop update steps.

## 3 `PERK`: META-LEARNING PARAMETER-EFFICIENT REASONING OVER KNOWLEDGE

We propose `PERK`, a bi-level optimization algorithm for model reasoning via test-time adaptation for any given long context. At inference time, `PERK` adapts a base model to encode a context via CLM optimization on that context. Subsequently, the adapted model can be used to answer relevant questions about that context. To scale `PERK`'s adaptation to larger and more capable base LLMs for effective long-context reasoning, we use parameter-efficient methods to reduce both the size and amount of gradient unrolling required during optimization.

### 3.1 PARAMETER-EFFICIENT META-LEARNING WITH LOW-RANK ADAPTATION

During training, `PERK` optimizes a parameter-efficient adapter, LoRA (Hu et al., 2022), instead of the complete model. Specifically, `PERK` encodes a context into the adapter through gradient-based updates, leaving the base model parameters unchanged. We denote by $\boldsymbol{\theta}_{\text{base}}$ and $\boldsymbol{\theta}_{\text{adapter}}$ the parameters of the base model and the LoRA adapter, respectively. $\boldsymbol{\theta}_{\text{base}}$ is never optimized, while $\boldsymbol{\theta}_{\text{adapter}}$ is meta-learned to a good initial state for test-time adaptations.

**Test-Time Learning and Inner Loop Training.** We set the test time learning objective (thus also the inner loop objective) to causal language modeling of the context: $\mathcal{L}_{\text{NLL}}(\mathcal{K}, (\boldsymbol{\theta}_{\text{base}}, \boldsymbol{\theta}_{\text{adapter}}))$, and set the adaptation algorithm $\mathcal{A}lg(\mathcal{L}_{\text{NLL}}, \mathcal{K}, (\boldsymbol{\theta}_{\text{base}}, \boldsymbol{\theta}_{\text{adapter}}), \boldsymbol{h})$ to only update $\boldsymbol{\theta}_{\text{adapter}}$. The adaptation algorithm computes the gradient $\nabla\mathcal{L}_{\text{NLL}}$ on a batch of sub-sequences from the full context sequence $\mathcal{K}$. This compression as a parallel *batch* (or multiple batches) allows us to process lengthy sequences beyond the model's native context window. The exact manner by which $\mathcal{K}$ is broken into batches is a hyperparameter of the algorithm, and for efficiency reasons, we set it to one batch during training.

**Outer Loop Training.** In the outer loop, the model learns to reason over the encoded information in the adapter (memory scratchpad) given a problem $\boldsymbol{q}$ without access to the long context $\mathcal{K}$ explicitly in the text space. We optimize the meta parameters $\boldsymbol{\theta}_{\text{adapter}}$ to learn from $R$, the corresponding distribution of reasoning problems. The optimal $\boldsymbol{\theta}_{\text{adapter}}$ minimize the expected reasoning loss of the adapter's adaptation to each reasoning problem $r = (\mathcal{K}, \boldsymbol{q}, y) \sim R$:

$$\boldsymbol{\theta}^*_{\text{adapter}} = \arg\min_{\boldsymbol{\theta}_{\text{adapter}}} \mathbb{E}_{r \sim R}\left[ \mathcal{L}_{\text{reason}}\Big(\boldsymbol{\theta}_{\text{base}}, \phi_{\text{adapter}}((\boldsymbol{\theta}_{\text{base}}, \boldsymbol{\theta}_{\text{adapter}}), \mathcal{K}), \{(\boldsymbol{q}, y)\}\Big)\right], \tag{3}$$

where $\phi_{\text{adapter}}((\boldsymbol{\theta}_{\text{base}}, \boldsymbol{\theta}_{\text{adapter}}), \mathcal{K}) = \mathcal{A}lg(\mathcal{L}_{\text{NLL}}, \mathcal{K}, (\boldsymbol{\theta}_{\text{base}}, \boldsymbol{\theta}_{\text{adapter}}), \boldsymbol{h})$ is the parameter-efficient CLM adaptation of $\boldsymbol{\theta}_{\text{adapter}}$ to the context $\mathcal{K}$ obtained via the test-time adaptation. We may use any optimizer for $\boldsymbol{\theta}_{\text{adapter}}$, and in our experiments, we use AdamW (Loshchilov & Hutter, 2017). We describe further hyperparameters, training & inference details, and long-context data preprocessing in Appendix B.

## 3.2 Scalable Meta-Learning with Truncated Gradient Unrolling

To optimize Equation (3) with gradient-based methods, we need to differentiate through $\mathcal{A}lg$, which involves higher-order derivatives and saving the complete trajectory to compute the meta-gradient (i.e., the gradient of $\mathcal{L}_{\text{reason}}$). However, this creates high memory costs, capping the method's applicability. To reduce memory costs, we truncate the backpropagation for the meta-gradient computation. We briefly describe this approach here, presenting the case where $\mathcal{A}lg$ optimizes $\boldsymbol{\theta}_{\text{adapter}}$ via direct gradient descent for simplicity (though we use more complicated optimizers for the inner loop in practice).

**Gradient Unrolling (GU).** We denote an N-step inner-loop optimization's intermediate states by:

$$\phi^{(0)}_{\text{adapter}} = \boldsymbol{\theta}_{\text{adapter}}, \quad \phi^{(n+1)}_{\text{adapter}} = \phi^{(n)}_{\text{adapter}} - \alpha\, g^{(n)}, \quad \phi^*_{\text{adapter}} = \phi^{(N)}_{\text{adapter}},$$

where $g^{(n)} = \nabla_{\phi^{(n)}_{\text{adapter}}} \mathcal{L}_{\text{NLL}}\big(\boldsymbol{\theta}_{\text{base}}, \phi^{(n)}_{\text{adapter}}, \mathcal{K}\big)$ is the gradient at step $n$, and $\alpha$ is the learning rate.

In vanilla MAML, to compute the gradient of the outer-loop loss $\mathcal{L}_{\text{reason}}$ with respect to the meta-parameters $\boldsymbol{\theta}_{\text{adapter}}$, one backpropagates through all $N$ inner steps. By the chain rule, this equates to accumulating the Jacobian $J^{(n)} = \frac{\partial \phi^{(n+1)}_{\text{adapter}}}{\partial \phi^{(n)}_{\text{adapter}}}$ from each step:

$$\nabla_{\boldsymbol{\theta}_{\text{adapter}}} \mathcal{L}_{\text{reason}} = \frac{\partial \mathcal{L}_{\text{reason}}}{\partial \phi^{(N)}_{\text{adapter}}} J^{(N-1)} J^{(N-2)} \cdots J^{(0)} \frac{\partial \phi^{(0)}_{\text{adapter}}}{\partial \boldsymbol{\theta}_{\text{adapter}}} = \frac{\partial \mathcal{L}_{\text{reason}}}{\partial \phi^{(N)}_{\text{adapter}}} \prod_{n=0}^{N-1} J^{(n)}, \tag{4}$$

where $J^{(n)}$ can be computed via: $J^{(n)} = I - \alpha H^{(n)}$, $H^{(n)} = \nabla^2_{\phi^{(n)}_{\text{adapter}}} \mathcal{L}_{\text{NLL}}$. Each Jacobian $J^{(n)}$ requires storing the computation graph of step $n$ in memory, so the cost of the backward pass scales linearly with $N$, creating a high memory requirement for gradient unrolling.

**Truncated Gradient Unrolling (TGU).** To reduce memory, we follow (Shaban et al., 2019) and run the inner loop for all $N$ specified update steps, but only store the computational graph for the last $T \ll N$ steps. This substantially reduces the computational memory requirements of the method, at the cost of slightly less accurate meta-gradients for the outer-loop optimization of $\boldsymbol{\theta}_{\text{adapter}}$. To implement this approximation, we do not retain the inner loop optimization's computation graph until the start of the *retention window*: the last $T$ inner-loop optimization steps. Since all Jacobians before step $n_{\text{start}}$ are treated as constants, the Jacobian product in Equation (4) is effectively cut:

$$\nabla_{\boldsymbol{\theta}_{\text{adapter}}} \mathcal{L}_{\text{reason}} \approx \frac{\partial \mathcal{L}_{\text{reason}}}{\partial \phi^{(N)}_{\text{adapter}}} \underbrace{\prod_{n=N-T}^{N-1} (J^{(n)})}_{\text{last } T \text{ steps}} \underbrace{\prod_{n=0}^{N-T-1} \cancel{(J^{(n)})}}. \tag{5}$$

## 4 Experiments

We empirically evaluate PERK's long-context reasoning performance in three scenarios: reasoning over (1) Needles-in-a-Haystack (NIAH, Kuratov et al., 2024), (2) Open-domain, multi-document QA (Multi-Doc), and (3) Drops-in-an-Ocean (DIO), a novel long context evaluation setting we propose.

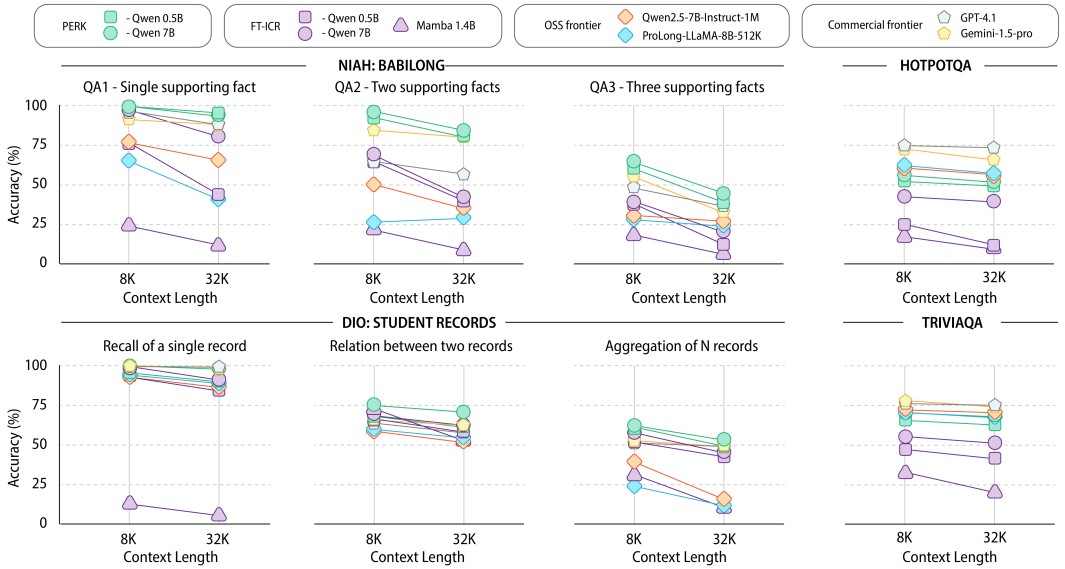

Figure 2: **Performance on Long-context Reasoning**. We show the evaluation results on NIAH with BabiLong, Multi-Doc with HotpotQA & TriviaQA, and DIO with Student Records. All `PERK` and FT-ICR models (including Mamba) are trained on contexts with 8K tokens. When evaluated on the out-of-distribution contexts with 32K tokens, they must extrapolate to a new context length. Note that we use a substring exact match as the Accuracy metric for HotpotQA & TriviaQA.

## 4.1 EXPERIMENTAL SETUP

**Datasets.** We use the BabiLong framework (Kuratov et al., 2024) for reasoning over Needles-in-a-Haystack (NIAH), with single (**QA1**), two-hop (**QA2**), and three-hop (**QA3**) reasoning over facts scattered across very long documents. We sample documents from FineWeb-Edu (Lozhkov et al., 2024) postdating our base model's knowledge cutoff. To address NIAH's fundamental limitation, where target information is stylistically distinct from distractors and thus artifically simpler to identify, we evaluate in two additional settings.

First, we evaluate on HotpotQA (Yang et al., 2018) and TriviaQA (Joshi et al., 2017) for real-world long-context QA. Following Yen et al. (2025), we inject retrieved distractors (Zhang et al., 2024a) at random positions until reaching the context window limit. We report substring exact match (SubEM) as accuracy following (Asai et al., 2023).

Second, we propose *Drops-in-the-Ocean* (DIO), a novel evaluation with distributionally similar relevant/irrelevant information. We create **Student Records**, a synthetic dataset where each context contains multiple student records (ID, name, school, major, grade). Context length scales with record count. We define three tasks: Recall (retrieve attributes by ID), Relation (compare attributes between IDs), and Aggregate (compute max/min/average grades). Test entities and attribute combinations are disjoint from training to ensure generalization. Examples are available in Appendix A.

**Baselines.** We compare `PERK` against multiple categories of baselines. First, following conventional practice (Fu et al., 2024; Gao et al., 2025b), we finetune open-weight models for prompting-based in-context reasoning. Our baselines include: (1) GPT-2-127M (Radford & Narasimhan, 2018) with 1K context window, extended via positional interpolation (Karypis et al., 2024) for longer contexts; (2) Qwen-2.5 (0.5B, 7B; Qwen et al., 2025) with 32K context; (3) LLaMA-3.2-1B and LLaMA-3.1-8B (Grattafiori et al., 2024) with 128K context; (4) Mamba-1.4B (Gu & Dao, 2024) with unlimited context length. We also compare to commercial models with long-context abilities, GPT-4.1 (OpenAI, 2025) and Gemini-1.5-Pro (Team et al., 2024), and specialized open-source models: Qwen2.5-7B-Instruct-1M (1M context via long-context pretraining, progressive length extension, and long-instruction post-training; Yang et al., 2025) and ProLong-Llama-3-8B-512k-Instruct (512K context; Gao et al., 2025a). We use 4-shot in-context learning as the inference method, following the prompt templates of Kuratov et al. (2024) and Yen et al. (2025)

## 4.2 Long-Context Reasoning Results

**NIAH.**  In the top row, left section of Figure 2, we show the evaluation results on BabiLong. `PERK` outperforms all baselines on all three of the **QA1**, **QA2**, and **QA3 tasks**, including much larger models and specialized long-context models that support much larger context-windows than 32K tokens (512K - 1M). When comparing similar training settings (*i.e.*, training on 8k contexts), `PERK` (Qwen-7B) achieves a higher accuracy across all task complexities than FT-ICR (Qwen-7B) with an in-distribution context window of 8K at evaluation time. When extrapolating to a 32K context window, `PERK` is much more robust, outperforming FT-ICR by 23% points on average.

**Multi-Doc.**  Next, we evaluate `PERK`'s performance on multi-document open-domain QA tasks HotpotQA and TriviaQA. When trained on long contexts with 8K tokens, `PERK` (Qwen) consistently outperforms FT-ICR (Qwen) for both HotpotQA and TriviaQA, achieving a 20% absolute gain at the 0.5B scale and 15% gain at the 7B scale. When extrapolating to contexts with 32K tokens, `PERK` shows stronger generalization over FT-ICR by 30% (0.5B) and 14% (7B). Compared to open-source long-context models (Qwen-1M and ProLong) with extensive training on long-context instruction data, `PERK` still matches the performance with a very narrow performance gap (3% for HotpotQA and 1.5% for TriviaQA). While not quite at the level of commercial LLMs, GPT-4.1 and Gemini-1.5-pro, `PERK`'s performance is closer to these leading performances than FT-ICR. Our results demonstrate `PERK`'s strong potential for retrieving from and reasoning over multiple long-context documents.

**Drops-in-the-Ocean.**  Results on the Student Records dataset in Figure 2 mirror the trends seen in BabiLong. `PERK` (Qwen) outperforms FT-ICR baselines across task difficulties and context lengths and shows a smaller drop than FT-ICR baselines when increasing the evaluation context length to 32k tokens. The performance differential between `PERK` and FT-ICR baselines also widens monotonically with task difficulty (Aggregate > Relation > Recall), suggesting that `PERK` enhances complex reasoning capabilities over long contexts rather than merely improving simple retrieval operations. Finally, as before, both `PERK` (Qwen) 7B and 0.5B outperform specialized long-context models despite their large parameter size and optimized training strategies for long sequences. `PERK` (Qwen-7B) also achieves performance that matches or exceeds the commercial models, Gemini-1.5-pro and GPT-4.1, even on the Aggregate task.

**Model Generalizability.**  We further analyze whether `PERK`'s performance can generalize across diverse model families and scales in Figure 3. While FT-ICR shows highly variable accuracy ranging from 18.2% on GPT-2 to 89.8% on LLaMA-8B, `PERK` maintains consistently high accuracy across all tested models. `PERK`'s performance scales positively with model size, achieving 4% increasing on Qwen-7B compared to Qwen-0.5B, and reaching 99.1% accuracy on LLaMA-8B. FT-ICR, despite improving with model size, still lags behind `PERK` by 9.3 percentage points even on the largest model tested. The generalization across model families is also clear. `PERK` achieves strong performance

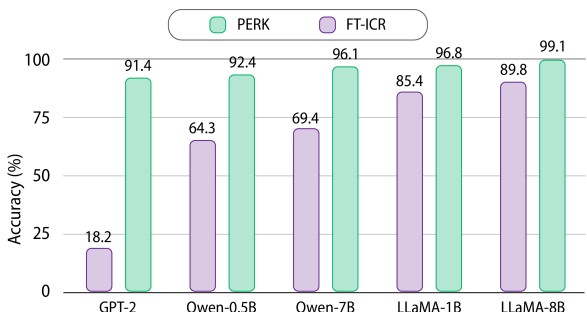

Figure 3: **`PERK` with diverse model scales and families** on BabiLong QA2 with an 8K train & test context length. `PERK`'s performance generalizes across models and scales, consistently outperforms FT-ICR baselines.

on models from the GPT-2, Qwen-2.5, and LLaMA-3.1/3.2 families. Our results demonstrate that `PERK`'s advantages are not limited to specific architectures or model scales, but rather represent a fundamentally effective approach for long-context modeling.

## 5 Analysis

In this section, we analyze `PERK`'s robustness in two challenging settings: test-time length robustness, where `PERK` is evaluated on longer (or shorter) contexts than those seen during long-context training (Press et al., 2022), and positional variation, where relevant contextual information is distributed throughout the long input context window (Liu et al., 2024).

## 5.1 TEST-TIME LENGTH ROBUSTNESS

**Setup.** Focusing on BabiLong tasks, which allow us to generate contexts of arbitrarily long lengths, we train a Qwen-2.5-0.5B base model using `PERK` on fixed-length contexts ranging from 1K to 8K tokens (as well as an FT-ICR baseline). Subsequently, we test on contexts ranging from 1K to 32K tokens (Figure 4). For analyzing length generalization on extreme context windows, we also extrapolate on contexts with 64K and 128K tokens (Table 1). We also compare `PERK` to recent training-free length generalization methods, Yarn (Peng et al., 2023b) and Dual Chunk Attention (DCA; An et al., 2024a) by applying these techniques to the FT-ICR Qwen model at inference time.

**PERK generalizes to new context lengths at test time.** Figure 4 shows that `PERK` outperforms FT-ICR on the BabiLong QA1 (single-hop) and QA2 (two-hop) tasks for all settings. FT-ICR's performance drops drastically as the inference context length exceeds the training context length, especially for shorter training contexts. While `PERK`'s accuracy also decreases (from a higher starting point), the degradation is dampened at longer test lengths. Meanwhile, for interpolation, FT-ICR performance drops on shorter contexts when finetuned on longer ones, matching observations from prior work (Gao et al., 2025b). In contrast, `PERK` maintains (and even increases) its performance.

To stress-test the limit of `PERK` models' length extrapolation ability, we test on sequences with context lengths beyond the context window, 32k, of the Qwen-2.5 model – 64K, and 128K tokens in Table 1, and demonstrate that `PERK` consistently extrapolates to extreme contexts lengths more robustly than long-context finetuning (FT-ICR). With 128K token context windows, `PERK`'s accuracy experiences a noticeable drop to 61.4% and 44.4% ac-

| Model | (a) Single-hop reasoning | | | (b) Two-hop reasoning | | |
|---|---|---|---|---|---|---|
| | 32K | 64K | 128K | 32K | 64K | 128K |
| *Commercial Frontier Models* | | | | | | |
| GPT-4.1 | 87.8 | 78.7 | 69.4 | 80.6 | **66.4** | **48.2** |
| Gemini-1.5-pro | 87.7 | 82.3 | **73.1** | 56.4 | 48.8 | 40.2 |
| *Trained on contexts with >256K tokens* | | | | | | |
| Qwen2.5-7B-Instruct-1M | 65.7 | 30.3 | 21.4 | 35.3 | 21.0 | 12.2 |
| ProLong-8B-Instruct-512K | 41.0 | 33.2 | 24.3 | 29.0 | 18.5 | 17.7 |
| *Trained on contexts with 8K tokens* | | | | | | |
| FT-ICR (Qwen2.5-0.5B) | 43.8 | 34.5 | 0 | 39.4 | 11.7 | 0 |
| w/ Yarn + DCA | 59.2 | 35.5 | 25.4 | 42.5 | 26.3 | 18.5 |
| `PERK` (Qwen2.5-0.5B) | **95.3** | **88.9** | 61.4 | **80.9** | 62.5 | 44.4 |

Table 1: **Test-time length extrapolation beyond 32K** on BabiLong QA1 (a) and QA2 (b). `PERK` and FT-ICR are trained on 8K-token sequences. The context length for inference grows from 64K to 128K. `PERK` extrapolates substantially better than FT-ICR.

curacy for QA1 and QA2, respectively, but these scores remain substantially better than the FT-ICR baseline at this same context length. After applying the training-free length generalization methods, Yarn and DCA, to FT-ICR Qwen models, the baselines show improved generalization on the extreme contexts. However, Yarn + DCA's extrapolation performance is still far below `PERK` for both QA1 and QA2 (around 35% lower on average). Compared to the specialized frontier models, `PERK` outperforms gemini-1.5-pro and the two open-source models, and achieves a narrow performance gap to GPT-4.1 on both 64K and 128K contexts. We note, as well, that `PERK` is only trained on contexts with 8K tokens, while the Qwen-1M and ProLong models have been trained on much longer contexts with 256K and 512K tokens. Overall, `PERK` demonstrates strong length generalization, extrapolating to longer contexts while preserving high performance on shorter ones.

## 5.2 ROBUSTNESS TO POSITIONAL BIASES

Prior work has shown that the position of relevant information in long contexts affects a language model's ability to use that information (Liu et al., 2024). In this experiment, we test `PERK`'s robustness to positional biases by evaluating its test-time performance on contexts where the position of relevant information is distributed differently from those seen in training.

**Setup.** We use API documents from Patil et al. (2024), where each document has an associated API call, and create long contexts containing multiple API-call-to-document pairs. In this task, the model must retrieve the correct API call based on user instructions and documents. Using fixed 4k and 8k token contexts, we train `PERK` and FT-ICR models on contexts with the target document at varying positions and test them on contexts distributed across all positions.

**PERK generalizes regardless of information position in the context.** Figure 5 compares `PERK` to FT-ICR under different position distributions (at train and test time) and context lengths. When trained on contexts with relevant documents randomly located (**Rnd**) throughout the context, `PERK`

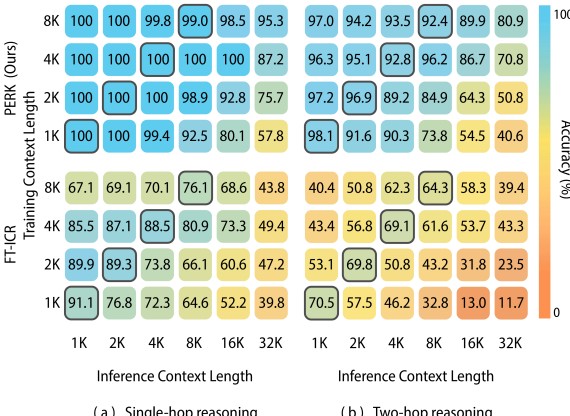

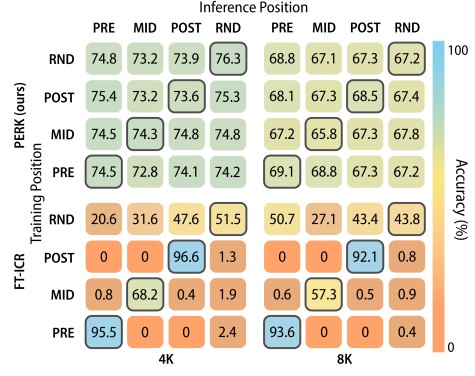

Figure 4: **Test-time context length robustness evaluation** on BabiLong QA1 (a) and QA2 (b) between PERK and FT-ICR on the Qwen-2.5-0.5B model. The y-axis represents the training context lengths, while the x-axis indicates various test-time context lengths. We test for both test lengths shorter than the training length, and test lengths longer than the training length. Bordered cells denote evaluation on context lengths equal to those in training. PERK shows stronger robustness across both settings.

Figure 5: **Positional Bias.** Comparison of PERK and FT-ICR on 4K and 8K contexts, on Qwen-2.5-0.5B. We train on problems where the relevant information appears in the beginning (**Pre**), middle (**Mid**), or end (**Post**) of the context, and evaluate on all three positional settings. We also train models on contexts where the relevant information is randomly located (**Rnd**), testing these on all four positional distributions (**Pre**, **Post**, **Mid**, **Rnd**). Bordered cells show in-distribution performances. PERK demonstrates strong positional robustness.

outperforms FT-ICR by 24%. Furthermore, FT-ICR shows large performance drops when the relevant document *consistently* appears at different locations (**Pre**, **Mid**, and **Post**) at test time. Meanwhile, shifting relevant positions at test-time shows minimal effect (within 1-2%) on PERK.

To further stress-test positional generalization, we also force relevant documents into specific positions during training and testing, spanning the full context. We find that FT-ICR easily overfits to the position pattern, completely failing to generalize to test-time position changes (performance drops to close to 0% when the position shifts at test time). Although PERK underperforms FT-ICR when the train-test positions are the same and the beginning (Pre) or end (Post) of the context, performance remains consistent regardless of the positional distribution shift between train and test time. We attribute PERK's robustness in this setting to the fact that documents in the long context are encoded as members of a permutation-invariant *batch* into the parameter space, limiting the overall impact of absolute position in the sequence. Meanwhile, FT-ICR directly aligns specific positions with answers during training.

## 5.3 INFERENCE LATENCY AND MEMORY USAGE

We evaluate PERK's efficiency on long contexts by measuring its inference memory cost and run-time, compared to in-context reasoning with finetuned models (FT-ICR).

**Setup** We complete all runs on a single Nvidia H100 GPU with 93 GB of VRAM and with Huggingface's Transformers library version 4.51.3. We perform four gradient update steps for the test-time inner loop adaptations for PERK. At inference time, all methods generate 64 tokens through greedy decoding using Huggingface's text generation pipeline. We measure runtime complexity in seconds (wall-clock time) and memory complexity in gigabytes. Both methods undergo 10 hardware warm-up iterations.

As explained in Section 3, PERK splits long sequences into batches of shorter segments. In this experiment, each batch has segments with an effective context length of 128 tokens. Since PERK performs gradient-based encoding over these batches, we can apply gradient accumulation to trade

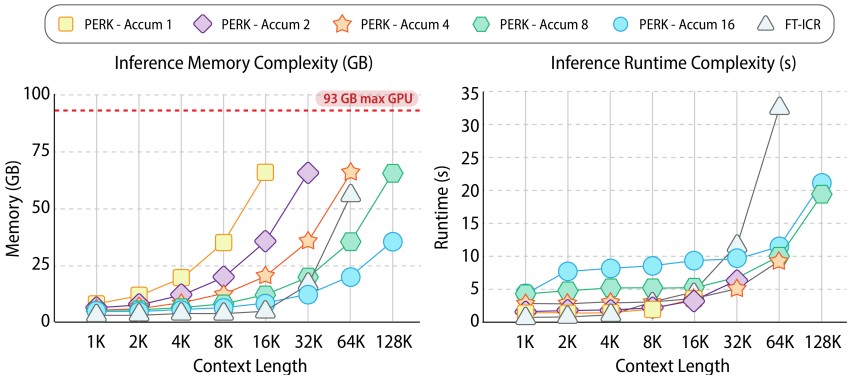

Figure 6: **Inference-time memory footprint and wall-clock runtime** as context length increases (up to 128K tokens), comparing PERK with FT-ICR. Curves that terminate before 128K indicate the method failed with an OOM error at longer context lengths, preventing further measurement. PERK demonstrates more efficient scaling in both memory and runtime, particularly for longer sequences.

runtime for reduced memory consumption. We thus analyze PERK's efficiency across varying gradient accumulation steps, ranging from 2 to 16.

**PERK scales more efficiently in memory and runtime on extremely long contexts.** Figure 6 shows that PERK provides more efficient memory and runtime scaling for extremely long contexts compared to FT-ICR. While FT-ICR is initially more efficient, its memory and runtime grow rapidly, leading to OOM errors at a context length of 128K. In contrast, PERK can manage the long sequences through gradient accumulation, which, while increasing runtime, reduces the memory footprint. For instance, at 8K tokens, increasing the accumulation steps from 1 to 16 reduces memory usage from 35.2GB to 5.9GB, although the runtime increases from 1.9s to 8.5s. Crucially, even with this runtime trade-off, PERK runs faster than FT-ICR at very long contexts where both can operate. At 64K tokens, FT-ICR takes 32.6s and 55.7GB, whereas PERK with 16 steps of accumulation completes in 11.4s, using only 19.6GB. With 8 steps, it is even faster at 9.7s, using 35.2 GB. Ultimately, at 128K tokens, where FT-ICR fails, PERK with 16 steps successfully processes the context using 35.2GB in 20.9s, showing that PERK provides a practical path to handle extreme context lengths efficiently in both memory and runtime compared to standard approaches.

## 6 RELATED WORK

**Test-Time Learning.** MAML (Finn et al., 2017) introduces optimization-based model-agnostic meta-learning for few-shot adaptation. Chen et al. (2023b) applies MAML to GPT models, integrating knowledge into model parameters through test-time gradient updates for downstream reasoning. Other methods achieve test-time learning in the model architecture. Sun et al. (2025) proposes RNN layers with learnable hidden states updated via self-supervised gradients on test inputs. Titans (Behrouz et al., 2024) incorporates a differentiable memory with transformers for rapid inference-time memorization. Building on Titan, ATLAS (Behrouz et al., 2025) proposes a new transformer architecture that contains the differentiable memory module as part of the network to support large-scale distributed pretraining. Other approaches adapt through context alone. MetaICL (Min et al., 2022) meta-learns from in-context examples without gradient updates. Our method, PERK, introduces parameter-efficient meta-learning using lightweight adapters, improving scalability for long contexts.

**Long-Context Language Models.** Developing long-context models has become a major research topic of its own. Many works explore extending transformer-based language models' context windows with minimal training, either by position extrapolation (Chen et al., 2023a; Peng et al., 2023c; Xiao et al., 2024; Yen et al., 2024) or improving the transformer's attention mechanism (Bertsch et al., 2023; Mohtashami & Jaggi, 2023; Shen et al., 2023; Jin et al., 2024). Other works show that using the default attention, applying simple position extrapolation, and fine-tuning the model on long documents (Fu et al., 2024; Lu et al., 2024) or synthetically-generated long data (An et al., 2024b; Xiong et al., 2024) yields much stronger results. The FT-ICR baseline in our work follows this paradigm. To reduce the expansive computation cost of long-context training, Chen et al. (2024)'s

work combines LoRA with trainable embedding and normalization to achieve efficient finetuning on long contexts. Other LLM works (Lieber et al., 2024; Grattafiori et al., 2024; Gao et al., 2025b) propose achieving long-context capabilities through a long-context continued training stage between standard pre-training and supervised fine-tuning. In contrast to all the approaches above, PERK achieves long-context reasoning through the ability to encode long sequences using test-time gradient updates on a lightweight model adapter.

**Long-Context Architectures.** In parallel, there have been many efforts in designing more efficient architectures, for example, linear attention with RNNs (Chandrayan et al., 2023; Peng et al., 2023a; Gu & Dao, 2024; Dao & Gu, 2024; Yang et al., 2024; Beck et al., 2024) and alternative attention architectures (Rubin & Berant, 2024; Sun et al., 2024; Han et al., 2023). However, these new architectures often require training from scratch, and many have inherent limitations in terms of long-context recall (Jelassi et al., 2024; Arora et al., 2025). Recent works also explore hybrid models (Lieber et al., 2024; De et al., 2024) or distilling existing LLMs into hybrid models (Wang et al., 2024) and show promising results. In comparison, PERK directly augments off-the-shelf pre-trained LLMs without requiring additional architecture modifications or extensive pre-training, while naturally being parameter-efficient.

**Long-Context Evaluation.** Many benchmarks have been proposed to evaluate language models' long-context abilities (Krishna et al., 2023; Shaham et al., 2023; Bai et al., 2024; Zhang et al., 2024b; Hsieh et al., 2024; Yen et al., 2025; Ye et al., 2025; Xia et al., 2025). Some works focus on the impact of extending input lengths with irrelevant contexts (Kuratov et al., 2024; Levy et al., 2024) or changing relevant information positions (Liu et al., 2024) on LLM reasoning. Many works also reframe natural-language-processing tasks, such as retrieval (Lee et al., 2024; Qiu et al., 2025), summarization (Kim et al., 2024), and in-context learning (Li et al., 2024; Xu et al., 2024; Bertsch et al., 2025), as long-context challenges. In this work, we define *Drops-in-the-Ocean* as a new long-context evaluation that makes distractors distributionally similar to relevant information.

## 7 CONCLUSION

In this work, we introduce PERK, a parameter-efficient meta-learning approach that learns to perform long-context reasoning by encoding information into a lightweight adapter through test-time gradient updates. Our experiments on multiple long-context benchmarks (Needle-in-the-Haystack, HotpotQA, TriviaQA, Drops-in-the-Ocean, and Code API retrieval) demonstrate substantial performance gains over models finetuned on long contexts for classical in-context reasoning across reasoning complexity and context lengths. We demonstrate that PERK's performance robustly transfers across model scales (0.5B to 8B) and model families (GPT-2, Qwen2.5, LLaMA-3). Our generalization analysis highlights PERK's strong test-time length extrapolation ability, where the model can generalize to unseen contexts $32\times$ longer than the training data. When relevant information shifts positions at test-time, PERK robustly generalizes across different positions. Overall, PERK excels in performance, generalization, and robustness for long-context reasoning.

### ACKNOWLEDGMENTS

We gratefully acknowledge the support of the Swiss National Science Foundation (No. 215390), Innosuisse (PFFS-21-29), the EPFL Center for Imaging, Sony Group Corporation, and a Meta LLM Evaluation Research Grant.

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

# A DATASETS

In this section, we provide details on the datasets used for each task in our experiments and analysis. For each dataset, we visualize an example in Figure 7.

## A.1 BABILONG

We construct the train and evaluation data using the data generation framework proposed by Kuratov et al. (2024). As we mentioned in the experiment setup, we mainly focus on tasks involving single-hop (**QA1**), two-hop (**QA2**), and three-hop (**QA3**) reasoning.

**Tasks**  The main task for BabiLong is to reason over facts distributed in a long context. For each QA task, we have a set of facts, including distractors and supporting facts. Each fact describes an agent's action or movement (which also describes the agent's location). For example, *Mary journeyed to the garden* is a movement, where *Mary* is the agent, and *garden* is Mary's current location. As an example for actions, *John picked up the apple there* describes John's action of picking up an apple. Together with multiple agents performing multiple movements and actions, the set of facts became a temporal sequence of events.

**QA1** focuses on one-hop reasoning, where the problems can be answered by using a single supporting fact. For example, in a sequence of actions and movements, the agent *John*'s last movement is *John moved to the bedroom*. Then the one-hop question is "*Where is John located now?*". By the last movement alone, we can find the answer "*bedroom*".

**QA2** focuses on one-hop reasoning, where the problems can be answered by using two supporting facts, where one fact is a movement and the other one is an action. For example, in the event sequence, we have *John moved to the garden* and *John dropped the apple there*. Using these two facts, we can answer "*Where is the apple now?*", which is "*garden*".

**QA3** focuses on one-hop reasoning, where the problems can be answered by using three supporting facts, as a combination of movements and actions. For example, we have three supporting facts: *Daniel went back to the office*, *Daniel went back to the bedroom*, and *Daniel left the milk* in order. We can infer the answer to the question "*Where was the milk before the bedroom?*" is "*office*".

**Build Long Context**  To build the long context, we use the framework to construct the tasks with long natural documents sampled from FineWeb-edu (Lozhkov et al., 2024), specifically from the *CC-MAIN-2024-42* subset, ensuring the content postdates our base model's knowledge cutoff. Given a particular context window, such as 8192, we sample that many tokens from the corpus as the *Haystack* long context.

We need to convert a long context into a batch of subsequences, such that `PERK` can encode them in parallel through test-time gradient-based adaptation. In practice, we set the effective context length (i.e., the number of tokens of a subsequence on average) as 256. We divide the long context into chunks where each chunk is approximately 256 tokens long. Then, we randomly distribute the set of facts into these chunks. For each fact, we randomly select a chunk and then insert this fact into the chunk at a random position. To preserve the temporal order of the facts (since they come from a sequence of events with temporal order), we index each fact with a prefix. Finally, we have a batch of subsequences where each sequence might contain some facts at random positions.

## A.2 STUDENT RECORDS

In the main paper, we propose *Drops-in-the-Ocean* (DIO), a new evaluation setting where long contexts are formed from structurally similar documents. We aim to create problems in which the relevant and irrelevant information will likely be distributionally similar, making it more difficult to identify critical facts. We construct a synthetic dataset, **Student Records**, where each context simulates a database containing multiple student records. Each record includes several attributes: ID, name, school, major, and grade.

**Tasks**  The task operates on top of a database that consists of multiple records. Each record holds information about a student. The record is structured as follows:

Student Id: 3109998, Student Name: Alison Keith, Year: Sophomore

School: School of Computer Information and Data Sciences, Major: Computer Science, Grade: 72

We generate the student IDs and grades using a random number generator. The student names are generated from the Name package of Python. We define a pool of years, schools, and courses. Then, we randomly select each entity for each student. Crucially, to test generalization, specific entities (e.g., IDs, names) and attribute value combinations encountered in the test set are disjoint from those used during training.

Based on the database of student records, we construct three main tasks: **Recall** (retrieving attributes for a specific ID), **Relation** (comparing attributes between two IDs), and **Aggregate** (calculating the maximum, minimum, and average grade across all students).

**Recall** tests a model's ability to retrieve an attribute of a student given a specific student ID number. For example, we can ask "*What major does student 1778924 study?*". The model should answer "*What major does student 1778924 study?*". The questions cover all attributes for testing coverage.

**Relation** tests a model's ability to compare the attributes of two students. For example, for the attribute of grade, we can ask "*Does student 1778924 have a higher grade than student 7077015?*". The model should be able to compare their grades and infer "*Yes*". Or we can also ask "*Do student 1778924 and student 7077015 study the same major?*", the model should recognize that their majors are different and answer "*No*".

**Aggregation** is the most difficult task in this dataset. This task requires many-hop reasoning ability, where the model needs to aggregate attributes from all students in a long context to make an accurate conclusion. We focus on three types of numerical aggregation among the students' grades: (1) Maximization, (2) Minimization, and (3) Averaging. For example, we ask the model "*What is the average grade of all students*", the model should be able to compute the sum of all student grades, count the total number of students in the context, and then compute the average, "*49.3*".

**Build Long Context**    Context length scales directly with the number of student records included. For a given context window, we continuously append student records to the context until the number of tokens exceeds the context window. To convert the long context into a batch of subsequences for PERK, we simply treat each student record as an individual subsequence in the batch, since they are already independent documents by design.

## A.3   OPEN-DOMAIN MULTI-DOC QA

We select HotpotQA (Yang et al., 2018) and TriviaQA (Joshi et al., 2017) as the main evaluation for open-domain QA over long-context documents. HoptpotQA tests the model's ability to perform 2-hop reasoning over two gold documents randomly located in a long list of documents. TriviaQA tests the model's ability to do reading comprehension over very long documents where the target information can be scattered around the long context. We refer to the original papers for QA construction details, data sources, and task examples of these two benchmarks.

**Build Long Context**    To build long contexts for these two benchmarks, given a specific context window (for example, 32768 tokens), we follow Yen et al. (2025)'s process for their RAG evaluation. We extend the original context by injecting retrieved distractor documents (Zhang et al., 2024a) at random positions until reaching the context window limit.

## A.4   LOST-IN-THE-MIDDLE API RETRIEVAL

To analyze a model's robustness against relevant information positions, we follow the setting proposed by *Lost in the Middle* (Liu et al., 2024). We use API documents from Patil et al. (2024), where each document has an associated API call, and create long contexts containing multiple API-call-to-document pairs. We place the relevant API document in a particular user instruction at different locations in the context (beginning, middle, end, and random position).

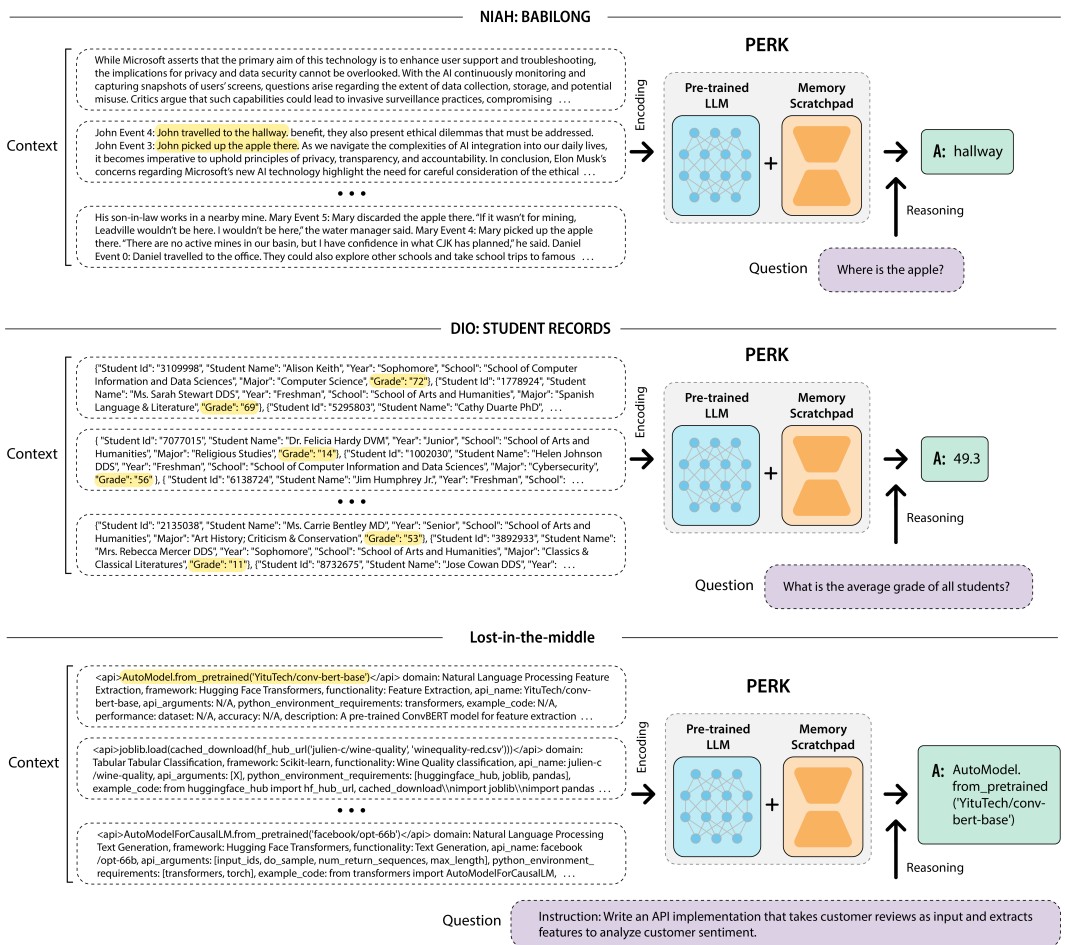

Figure 7: **Dataset Examples:** Here we show an example from the three datasets we used: (Top) BabiLong, (Middle) Student Records, and (Bottom) Lost-in-the-Middle API Retrieval. PERK first encode the contexts into the memory scratchpad through self-supervised test-time learning. Next, the user can ask any number of questions about that context, and the LLM can reason over the updated memory scratchpad to generate the answers.

**Task** We formulate the main task as gold API retrieval. In this task, the model must retrieve the correct API call based on user instructions and documents. Using fixed 4k and 8k token contexts, we train PERK and FT-ICR models on contexts with the target document at varying positions and test them on contexts distributed across all positions.

**Build Long Context** We follow Student Records and scale the context length with the number of API documents included. For a given context window, we continuously append API documents to the context until the number of tokens exceeds the context window. Similarly, since each API document is independent, we treat a document as a subsequence in the batch for PERK. Since the documents are processed in parallel as a batch, they do not preserve any ordering information. Thus, by design, PERK is robust to the position of the relevant information in a context.

## B  TRAINING AND INFERENCE DETAILS

**Hardware Information**   All training runs are conducted on a single Nvidia H100 GPU with 93GB of VRAM. The training jobs run on an internal research cluster managed by Slurm. Each job is provided with 64 CPUs and 64GB of RAM. The GPU is completely utilized without sharing with other concurrent jobs.

**Standard Hyperparameters**   Each training (for PERK and baselines) sets 2 epochs as the maximum runtime with early stopping based on validation loss. In practice, most training runs typically stop early, within one epoch of runtime. We set the baseline training and PERK's outer loop learning rate to 1e-5, and the weight decay to 0.01. We use a cosine learning rate scheduler with 0.03 of the total optimization steps as the warm-up phase. We use PyTorch Lightning [1] as the trainer for PERK. The Qwen and GPT-2 FT-ICR baselines are trained using the Open-Instruct (Lambert et al., 2025) trainer from AI2, and the Mamba FT-ICR baselines are trained using the Huggingface Trainer [2]. For the stress test of length generalization beyond 64K tokens, we change the default maximum position of 32768 to 131072.

**LoRA Hyperparameters**   We set LoRA rank to 256 for both PERK GPT-2 and PERK Qwen. We use the default values for the other hyperparameters, including setting the alpha to 16 and the dropout to 0.1. We apply LoRA to all modules in the model architecture. For 1B scale and below, we apply LoRA to all layers in the model. However, for 7B and 8B models, we apply it only to the top 4 layers in order to reduce the memory overhead of meta-learning. We set the scaling factor $\alpha$ to 16 initially and enable rank-stabilized LoRA fine-tuning Kalajdzievski (2023), which results in a target $\alpha$ of 256, corresponding to a rank of 256.

**Meta-Learning Hyperparameters**   For the inner loop adaptation, we use 4 inner loop gradient steps. For the truncated gradient unrolling, we truncate steps 1 and 2 for input context length $\leq$4K, and we truncate steps 1, 2, and 3 for context length $>$ 8K. We follow Ye & Chao (2022)'s practice on how to train a good MAML model and meta-learn a set of per-layer-per-step dynamic learning rates for the inner loop as part of the outer loop learning, using 5e-5 as the initial value. This approach learns a separate learning rate for each network layer and each adaptation step in the inner loop. These learning rates are optimized during the meta-training process alongside the model parameters. Note that these per-layer-per-step learning rates are learned parameters, not hyperparameters that require manual tuning. During meta-training, our algorithm automatically learns to adjust these learning rates for each layer and inner loop step dynamically based on the meta-objective. This approach reduces the hyperparameter tuning burden compared to manually setting the adaptive learning rates. For the language modeling loss in the inner loop, we use a weighted version similar to CaMeLS (Hu et al., 2023). The token-level weights are predicted via a weighting model implemented as a two-layer MLP network. The parameters of the weighting model are co-meta-learned along with the LoRA parameters and the adaptive per-layer-per-step learning rates. We use the differentiable version of the AdamW (Loshchilov & Hutter, 2017) optimizer for the inner loop optimization, implemented by the Higher [3] library for higher-order optimization. We compute higher-order derivatives using the automatic differentiation mechanism implemented by PyTorch[4]. The outer loop optimizer is the standard PyTorch implementation of AdamW.

**Long-context Data Collation for PERK**   At training time, to collate a reasoning problem with a long context and a set of associated question-answer pairs, we first convert a long context into a batch of subsequences, with relatively even lengths. In practice, we set the effective context length (i.e., the number of tokens of a subsequence on average) as 256, and divide the long context into $N$ chunks where each chunk is approximately 256 tokens long. For a long context with 8192 tokens, the batch size is thus $N = 32$. PERK's inner loop adaptation encodes the batch of subsequences in parallel through gradient updates. At inference time, if we want to test PERK on a long context with 32K

---

[1] https://lightning.ai/docs/pytorch/stable/
[2] https://huggingface.co/docs/transformers/en/main_classes/trainer
[3] https://github.com/facebookresearch/higher
[4] https://docs.pytorch.org/tutorials/beginner/blitz/autograd_tutorial.html

tokens, the effective context length stays the same as approximately 256 tokens, while the batch size increases to $N = 128$.

Note that splitting a long context into a batch of shorter sequences can either be permutation-invariant or maintain the original ordering. PERK induces ordering through explicit indexing in the prompt formatting, which allows the model to learn implicit ordering during the encoding phase. Specifically, when chunking a long context, we can preserve ordering information by indexing each chunk. For example, if we split a long document into segments *A, B, C, D, E*, we format them as: [*"Document 1: A", "Document 2: B", ..., "Document 5: E"*]. During the inner loop adaptation with the NLL objective, the LoRA parameters learn to associate these indices with their content, effectively encoding the ordering information.

**Evaluation Hyperparameters** At inference time, we generate the answer for a reasoning problem via greedy decoding (temperature is set to 0). The maximum number of tokens that can be generated is 512. The EOS token is set to the default one defined in the tokenizer. The generation is handled by the text-generation pipeline implemented by Huggingface [5]. For the length generalization analysis, we apply recent training-free methods Yarn (Peng et al., 2023b) and Dual Chunk Attention (DCA, An et al. (2024a)) to the finetuned Qwen models. We use vLLM [6] (0.10.2) as the inference engine to apply Yarn + DCA. For Yarn, we apply a factor of 4.0 to the original max position embeddings, 32768, and scale the context window to 131072. For DCA, we use vLLM's *DUAL_CHUNK_FLASH_ATTN* backend for generation.

## C    ADDITIONAL ANALYSIS

### C.1    IMPACT OF LoRA RANK IN PERK

In this section, we analyze the impact of LoRA's rank on PERK's long-context reasoning performance. We focus on the BabiLong tasks with a 1024 context window for an efficient ablation study. We apply PERK to Qwen-2.5-0.5B with LoRA rank of 16 and 256 and evaluate on the BabiLong tasks.

| Method | QA1 | QA2 | QA3 |
|---|---|---|---|
| FT-ICR | 91.1 | 70.5 | 48.6 |
| PERK LoRA-16 | 99.0 | 96.9 | 84.7 |
| PERK LoRA-256 | **100** | **98.1** | **89.1** |

Table 2: Analyzing LoRA rank's impact on PERK's performance.

We show the results in Table 2. We also include the performance from the FT-ICR Qwen baseline as a reference. The results show that PERK's performance only decreases slightly when the rank is reduced to 16. However, PERK still maintains a much stronger performance than the FT-ICR Qwen baseline. The analysis shows that PERK can maintain its strong performance with a more parameter-efficient setting, highlighting its potential for more efficient long-context reasoning.

### C.2    INNER LOOP ADAPTATION STEPS

In this section, we analyze the impact of inner loop adaptation steps on PERK's long-context reasoning performance. Similarly, we focus on the BabiLong tasks with a 1024 context window. We again apply PERK to Qwen-2.5-0.5B for 2 and 4 inner loop steps. With 4 steps, we truncate the first 2 steps in backpropagation. With 2 steps, we truncate the first step in backpropagation. We

| Method | QA1 | QA2 | QA3 |
|---|---|---|---|
| FT-ICR | 91.1 | 70.5 | 48.6 |
| PERK 2 Steps | 99.2 | 94.5 | 82.4 |
| PERK 4 Steps | **100** | **98.1** | **89.1** |

Table 3: Analyzing the impact of the number of inner loop adaptation steps on PERK's performance.

show the results in Table 3. The results show that reducing the number of inner loop adaptation steps decreases the performance more noticeably than reducing LoRA rank. However, PERK still outperforms FT-ICR Qwen on all three tasks. The results here suggest that we should select a higher number of inner loop adaptation steps to maximize the performance. If in settings where GPU memory might be limited, reducing the number of inner loop steps can reduce the performance, but it

[5] https://huggingface.co/docs/transformers/v4.43.4/en/main_classes/pipelines

[6] https://docs.vllm.ai/en/latest

is still able to outperform the baseline. However, the analysis is done with limited hardware resources, which limits our analysis to 4 steps. To further analyze the impact of the inner loop step counts, more systematic experiments in a scalable hardware setting are needed for future work.

### C.3    GRADIENT UNROLLING TRUNCATION STEPS

In this section, we analyze the impact of the truncation steps we use when performing gradient unrolling (GU) for computing the meta-gradients in the outer loop of the meta-learning process. Recall that to reduce the memory cost of full gradient unrolling, which requires saving the computation graph for each inner loop adaptation step, we truncate the unrolling by detaching the early steps' graph to save memory. Bounded by memory constraints, we can only train on 1K-token contexts with full GU without any truncation. Thus, we conduct our analysis under the 2K context window. Given 4 inner loop adaptation steps, we truncate 0-4

| Truncation Length | Accuracy |
|---|---|
| TGU (0) | 100 |
| TGU (1) | 100 |
| TGU (2) | 100 |
| TGU (3) | 98.8 |
| TGU (4) | 92.4 |

Table 4: Truncation length ablations on the BA-BILong QA1 task with 1K tokens. We conduct 4 steps of inner loop adaptation and truncate 0–4 steps. PERK maintains the accuracy up to 3 truncation steps, but drops when all steps are truncated.

steps with Qwen2.5-0.5B on the BabiLong-QA1 task. The analysis shows that we can truncate up to 3 steps and only unroll the last inner loop adaptation while maintaining a high accuracy without significant degradation. However, truncating all steps, which results in a first-order approximation of the meta-gradients, shows a large performance drop, even on this very simple setting with QA1-1K.

## D    TRAINING MEMORY COST ANALYSIS

We show PERK's training scalability by measuring its peak memory use across different context lengths (compared to a prior test-time learning method, RECKONING; Chen et al., 2023b).

**Setup** We complete all runs on a single Nvidia H100 GPU with 93 GB of VRAM using Huggingface's Transformers library version 4.51.3. We perform four gradient update steps for the training-time inner loop adaptations for PERK. We test four truncation levels (§3.2): truncate 0 (full gradient unrolling without truncation), 1, 2, and 3 steps. Both methods undergo 10 hardware warm-up iterations.

**PERK is more memory efficient at training.** Figure 8a shows that, as the training context length grows from 1K to 8K tokens, RECKONING quickly runs into Out-Of-Memory (OOM) errors at context length 2K. At the same time, PERK, with increasing truncation steps, continues to fit the maximum GPU memory, successfully validating PERK's training scalability on long contexts.

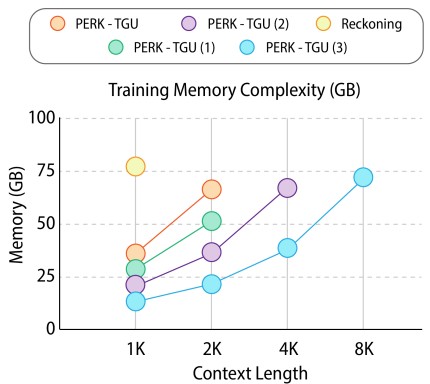

Figure 8: **Peak GPU memory usage during training** with context lengths ranging from 1K to 8K tokens for RECKONING and PERK. While PERK successfully scales to 8K tokens, RECKONING encounters out-of-memory (OOM) errors at shorter context lengths.

## E    DETAILED RESULTS

We show detailed quantitative performance of PERK, FT-ICR baselines, and frontier long-context LLMs on the long-context reasoning benchmarks we evaluate on in Table 5.

| Model | BBL-QA1 | | BBL-QA2 | | BBL-QA3 | | HotpotQA | | TriviaQA | | SR-Recall | | SR-Relation | | SR-Aggregate | | AVG | |
|---|---|---|---|---|---|---|---|---|---|---|---|---|---|---|---|---|---|---|
| | 8K | 32K | 8K | 32K | 8K | 32K | 8K | 32K | 8K | 32K | 8K | 32K | 8K | 32K | 8K | 32K | 8K | 32K |
| *Commercial Frontier Models* | | | | | | | | | | | | | | | | | | |
| Gemini-1.5-pro | 95.6 | 87.7 | 64.0 | 56.4 | 49.4 | 36.7 | **75.3** | **73.4** | 76.4 | **75.1** | **100** | **99.4** | 63.3 | 58.8 | 50.9 | 48.7 | 71.7 | 67.0 |
| GPT-4.1 | 91.4 | 87.8 | 84.5 | 80.6 | 54.9 | 33.1 | 72.4 | 65.8 | **78.3** | 74.3 | **100** | 98.8 | 66.5 | 62.5 | 52.4 | 49.4 | 75.1 | 69.0 |
| *Trained on contexts with >256K tokens* | | | | | | | | | | | | | | | | | | |
| Qwen2.5-7B-1M | 77.4 | 65.7 | 50.2 | 35.3 | 30.5 | 27.2 | 60.3 | 55.8 | 72.8 | 70.9 | 92.7 | 86.7 | 58.2 | 52.1 | 39.4 | 15.8 | 60.2 | 51.2 |
| ProLong-8B-512K | 65.3 | 41.0 | 26.2 | 29.0 | 27.8 | 24.4 | 62.4 | 57.3 | 70.4 | 68.2 | 94.5 | 89.2 | 60.1 | 54.4 | 24.3 | 12.1 | 53.9 | 47.0 |
| *Trained on contexts with 8K tokens* | | | | | | | | | | | | | | | | | | |
| FT-ICR (Mamba-1.4B) | 24.2 | 12.4 | 22.2 | 8.5 | 18.5 | 6.8 | 17.2 | 9.8 | 32.1 | 20.5 | 12.1 | 5.4 | 73.0 | 52.3 | 30.5 | 10.9 | 28.7 | 15.8 |
| FT-ICR (Qwen2.5-0.5B) | 76.1 | 43.8 | 64.3 | 39.4 | 37.5 | 12.1 | 24.4 | 11.9 | 47.4 | 41.7 | 93.6 | 84.5 | 65.3 | 57.8 | 51.8 | 42.4 | 57.6 | 41.7 |
| FT-ICR (Qwen2.5-7B) | 97.5 | 80.9 | 69.4 | 42.7 | 39.4 | 20.8 | 42.4 | 39.8 | 55.4 | 51.7 | 98.8 | 91.5 | 69.7 | 62.4 | 57.6 | 45.8 | 66.3 | 54.5 |
| *PERK (Ours), trained to encode contexts with 8K tokens* | | | | | | | | | | | | | | | | | | |
| PERK (Qwen2.5-0.5B) | 99.0 | **95.3** | 92.4 | 80.9 | 60.4 | 38.7 | 52.2 | 49.4 | 65.5 | 62.8 | 95.8 | 90.4 | 67.9 | 60.9 | 60.1 | 49.5 | 74.2 | 66.0 |
| PERK (Qwen2.5-7B) | **100** | 94.2 | **96.1** | **84.4** | **65.2** | **44.5** | 55.6 | 52.2 | 70.8 | 69.4 | **100** | 98.5 | **75.5** | **70.3** | **62.4** | **53.3** | **78.2** | **70.9** |

Table 5: **Model Performance Across Long-context Reasoning Benchmarks and Context Lengths (8K, 32K)**: We list detailed performance of our method PERK, FT-ICR baselines, and the frontier specialized long-context LLMs: Qwen2.5-7B-Instruct-1M (Yang et al., 2025), ProLong-LLaMA3-8B-Instruct-512K (Gao et al., 2025a), GPT-4.1 (OpenAI, 2025), Gemini-1.5-pro (Team et al., 2024). **BBL** here stands or BabiLong (Kuratov et al., 2024). SR stands for Student Records, our proposed dataset for Drops-in-the-Ocean reasoning. PERK and FT-ICR models have only seen training data contexts with 8K tokens, requiring them to generalize to the unseen 32K context window. PERK achieves the highest performance on average for both 8K-token and 32K-token contexts.

