# OpenReview forum: "PERK: Long-Context Reasoning as Parameter-Efficient Test-Time Learning"
_ICLR.cc/2026/Conference — ICLR 2026 Poster_

### Official Review · Reviewer_MTjy · 2025-10-15

**Soundness:** 3
**Presentation:** 4
**Contribution:** 3
**Rating:** 8
**Confidence:** 3

**Summary:**

This paper introduces PERK (Parameter Efficient Reasoning over Knowledge), a novel method that reframes long-context reasoning as a test-time learning problem. Instead of processing a long context in a single forward pass, PERK first encodes the context into the parameters of a Low-Rank Adapter (LoRA) via several gradient-based update steps at inference time. The model then answers a given question using this newly created parameter-efficient memory module, without needing the original long context in its attention window. To enable this, PERK is trained using a bi-level meta-learning optimization scheme: an inner loop learns to rapidly encode context into the LoRA adapter, while an outer loop learns to use the adapted model for reasoning. To make this process scalable, the authors employ truncated gradient unrolling. The paper demonstrates through extensive experiments that PERK significantly outperforms standard long-context fine-tuning, shows superior length extrapolation and robustness to positional biases, and is competitive with or exceeds the performance of specialized proprietary and open-source long-context models.

**Strengths:**

**Novel Conceptual Framing:** The paper's most significant contribution is reframing long-context reasoning as a test-time, gradient-based adaptation problem. This is a fundamental and clever departure from the standard in-context reasoning (ICR) paradigm, moving from "reasoning over context" to "reasoning from context stored in parameters."

**Exceptional Empirical Performance and Robustness:** PERK demonstrates substantial performance gains (up to +20% absolute) over standard fine-tuning. More importantly, the method shows remarkable robustness where standard models fail:
- Length Extrapolation: PERK models trained on 8K contexts maintain strong performance on contexts up to 128K, significantly outperforming baselines and even specialized long-context models.

- Positional Robustness: By processing context as a permutation-invariant batch of segments, PERK largely overcomes the "lost in the middle" problem, showing minimal performance degradation when the position of relevant information changes, unlike FT-ICR models which fail catastrophically.

**Scalability and Model-Agnosticism:** The combination of LoRA and truncated gradient unrolling makes the complex meta-learning approach tractable . The paper convincingly shows this advantage holds across multiple model families (GPT-2, Qwen, LLaMA) and scales (0.5B to 8B), highlighting the general applicability of the technique.

**Strong Evaluation Methodology:** The introduction of the Drops-in-the-Ocean (DIO) benchmark is a valuable contribution, creating a more challenging and realistic test than standard Needle-in-a-Haystack by using distributionally similar distractors.

**Weaknesses:**

**Inference Latency Trade-Off:** The paper emphasizes efficiency in terms of training memory, but the trade-off with inference latency is a major practical concern. The method requires performing multiple gradient updates at test time to encode the context before answering a question. While Figure 7b shows PERK can be faster at extreme context lengths (e.g., 64K), it appears slower than FT-ICR for more common lengths. This makes the "efficient" framing potentially misleading for many real-world applications where inference speed is critical.

**Complexity of the Training Framework:** The PERK training pipeline is highly complex, involving bi-level optimization, truncated gradient unrolling, learned per-layer-per-step learning rates, and a weighted NLL loss in the inner loop. This high degree of complexity could pose a significant barrier to reproducibility and adoption compared to the much simpler standard fine-tuning approach.

**Sub-Optimal Encoding Objective:** The inner loop's objective is simply next-token prediction over the context segments. This seems to incentivize the model to learn the surface-level statistics of the text, which may not be the most effective way to distill salient, reasoning-critical information into the LoRA parameters. The paper does not explore or discuss alternative, potentially more targeted, encoding objectives.

**Questions:**

1. Inference Latency Trade-Off: Could you provide a more direct comparison of the end-to-end inference latency (context encoding + question answering) of PERK vs. FT-ICR, particularly in the common 8K-32K context range? How does this latency scale with the number of inner-loop adaptation steps, and where is the practical break-even point where PERK becomes faster than a standard forward pass?

2. Optimality of NLL for Encoding: The inner loop uses a standard NLL objective to "compress" the context into the LoRA. Have you considered alternative objectives that might more explicitly encourage the adapter to store salient facts? For instance, would a contrastive loss or an information-theoretic objective that prioritizes reasoning-relevant information lead to a more effective parametric memory?

3. Comparison to RAG: Your method encodes the entire context into parameters, which is a key difference from RAG systems that first retrieve a small subset of the context. Could you discuss the conceptual trade-offs between PERK and a standard RAG pipeline in terms of performance, latency, and robustness to different types of reasoning tasks?

4. Ablation on Training Components: The PERK training process involves several advanced components (TGU, learned per-layer LR, weighted NLL). How critical is each of these to the final performance? For example, what is the performance drop if you use a fixed inner-loop learning rate or a standard (unweighted) NLL loss?

---

> ### Author Response · Authors · 2025-11-24
> **Response to Reviewer MTjy**
>
> We thank the reviewer for acknowledging the novelty of our framing of long-context reasoning as test-time learning. We appreciate that the reviewer recognizes PERK’s empirical improvements and robustness, its scalability and model-agnosticism, and our strong evaluation methodology. Below, we address their concerns and questions:
>
> ### **W1 & Q1: The reviewer is interested in the inference cost of PERK vs. FT-ICR**
> Our analysis in Appendix C (original version) and Section 5.3 (revised manuscript) provides measurements including both memory and runtime efficiency at inference time. While PERK does increase latency at shorter lengths, the break-even point for runtime occurs around 4K-8K tokens, where PERK's more efficient scaling begins to compensate for the gradient update overhead. This gradient update is responsible for the vast majority of the inference latency, and increasing the number of inner loop steps (currently 4) would roughly linearly increase the overall inference runtime.
>
> Broadly, PERK’s inference consists of a trade-off between memory and runtime efficiency. As the context length grows, we can increase the number of gradient accumulation steps to break the memory bottleneck. Runtime will increase, while memory will decrease, allowing us to model much longer context lengths than FT-ICR. When dealing with shorter context lengths, which FT-ICR manages with no problems, we can lower the number of gradient accumulation steps that PERK uses since the memory requirements are not that high, and speed up the runtime. We see this in Figure 7b (originally) or Figure 6b (revised manuscript), where PERK’s runtime with a small number of accumulation steps is close to or lower than that of FT-ICR. In fact, PERK only becomes slower than FT-ICR when context lengths are shorter than 4K.
>
> Overall, we believe the cost at these lower lengths remains justified by the advantages demonstrated in our paper, including (1) Substantial performance gains across all context lengths, including the 8K-32K range, (2) Robust length extrapolation, and (3) Positional robustness.

---

> > ### Author Response · Authors · 2025-11-24
> >
> > ### **W2 & Q4: The reviewer is interested in ablation studies on PERK’s algorithmic design**
> > In Appendix D of the original submission (Appendix C of the updated manuscript), we provide a detailed analysis of important design choices of PERK, including the rank of the LoRA adapter, the number of inner loop adaptation steps, and the number of truncation steps for TGU.
> >
> > Regarding learnable learning rates, they are a crucial component for the convergence of the meta-training of PERK. If a fixed learning rate is used in the inner loop, the training fails to converge. Thus, the learnable learning rate has been a default setting for the meta-learning algorithm throughout the development and experiments of this work.
> >
> > For the weighted loss, we conduct an additional ablation study using Qwen-2.5-0.5B and BabiLong’s QA2 task with 1K length. Our evaluation shows that PERK without weighted loss achieves 83.9% while PERK achieves 98.1%, echoing prior work [1] that the weighted loss is an important design decision.
> >
> > #### **Reference**
> > [1] Hu, N., Mitchell, E., Manning, C. D., & Finn, C. (2023). Meta-learning online adaptation of language models. arXiv preprint arXiv:2305.15076.

---

> > > ### Author Response · Authors · 2025-11-24
> > >
> > > ### **W3 & Q2: The reviewer is interested in alternative encoding objectives beyond NLL**
> > > Our experiments show that even with the canonical self-supervised objective (NLL), PERK achieves substantial improvements over the common FT-ICR, so we did not explore other inner loop training objectives.
> > >
> > > That said, we agree that other objectives could potentially improve PERK. We are excited about several directions, such as using the Predictive Information Bottleneck [1], which could explicitly balance encoding sufficient information for reasoning while maintaining compact representations. Approaches that align the encoded representations with explicit knowledge structures, such as FF-KV memory alignment [2] or logit-lens objectives [3], could also make reasoning-critical facts more directly accessible during the outer-loop reasoning phase.
> > >
> > > We view alternative encoding objectives as a valuable direction for future work. Having established that test-time learning via parameter adaptation is a promising paradigm, we look forward to the community building upon this foundation with more sophisticated encoding mechanisms.
> > >
> > > #### **Reference**
> > > [1] Alemi, A. A., Fischer, I., Dillon, J. V., & Murphy, K. (2016). Deep variational information bottleneck. arXiv preprint arXiv:1612.00410.
> > >
> > > [2] Mor Geva, Roei Schuster, Jonathan Berant, and Omer Levy. 2021. Transformer Feed-Forward Layers Are Key-Value Memories. In Proceedings of the 2021 Conference on Empirical Methods in Natural Language Processing, pages 5484–5495, Online and Punta Cana, Dominican Republic. Association for Computational Linguistics.
> > >
> > > [3] Belrose, N., Furman, Z., Smith, L., Halawi, D., Ostrovsky, I., McKinney, L., ... & Steinhardt, J. (2023). Eliciting latent predictions from transformers with the tuned lens. arXiv preprint arXiv:2303.08112.

---

> > > > ### Author Response · Authors · 2025-11-24
> > > >
> > > > ### **Q3: The reviewer is interested in the comparison between PERK and RAG**
> > > > PERK and RAG address fundamentally different stages of the long-context reasoning pipeline. RAG retrieves a small subset of relevant passages from a large corpus, then performs reasoning over the retrieved context using standard in-context learning. PERK compresses an entire given context into learned parametric representations useful for downstream reasoning through test-time adaptation, then performs reasoning over these representations.
> > > > RAG has the limitation of potentially selecting incorrect documents. In fact, for BabiLong’s QA2 and QA3 tasks, prior work [1] has shown that retrieval fails dramatically, with accuracy dropping below random guessing, even with a powerful model like GPT-4 (42% at 32K for QA2 and 31% at 32K for QA3). Meanwhile, PERK (Qwen2.5-0.5-B) achieves higher accuracy 80.9% at 32K for QA2 and 38.7% at 32K for QA3.
> > > >
> > > > However, we ultimately view these approaches as complementary rather than competing. A two-stage pipeline could leverage both strengths:
> > > >
> > > > 1. Stage 1 (Retrieval): Retrieve all potentially relevant documents from a large corpus.
> > > > 2. Stage 2 (PERK): Encode the retrieved documents into robust parametric memory for complex reasoning
> > > >
> > > > With this combination, the retrieving pipeline for RAG can help retrieve a larger set of documents. At the same time, PERK can perform more effective encoding for downstream reasoning than standard in-context reasoning in a RAG pipeline. We are interested in exploring this integration in future work.
> > > >
> > > > #### **Reference**
> > > > [1]  Kuratov, Y., Bulatov, A., Anokhin, P., Rodkin, I., Sorokin, D., Sorokin, A., & Burtsev, M. (2024). Babilong: Testing the limits of llms with long context reasoning-in-a-haystack. Advances in Neural Information Processing Systems, 37, 106519-106554.

---

### Official Review · Reviewer_sFKq · 2025-10-26

**Soundness:** 3
**Presentation:** 3
**Contribution:** 3
**Rating:** 4
**Confidence:** 3

**Summary:**

This work proposes a meta-learning method that learns to encode long contexts through gradient updates to a model adapter at test time. The inner encodes the context to Lora, and the outer loop uses the adapter for the relevant information from the encoded long-context. The experiment proves that the proposed PERK is effective for long-context reasoning.

**Strengths:**

* The test-time learning problem is relatively useful to further improve the model performance.
* Figure 1 presents the pipeline of this method, and the inner loop is clear.
* The method compares the performance with FT-ICR, proving that the PERK is better.
* The experiments have various models, supporting that the PERK is general.

**Weaknesses:**

* Is the method test-time training? According to the definition of the outer loop, Equation 3 needs the label of the question. However, during test time, there is no such label.
* Figure 4 is not related to length extrapolation. The Qwen-2.5-0.5B is trained with 32K by the Qwen Team. However, Figure 4 presents the performance with a maximum 32K, which is NOT longer than the Qwen-2.5-0.5B training length. This is a misclaim.
* It should provide the training cost, such as time cost, for the comparison of PERK and FT-ICR.

**Questions:**

N/A

---

> ### Author Response · Authors · 2025-11-24
> **Response to Reviewer sFKq**
>
> We thank the reviewer for recognizing the performance gain and generalizability of our method, the clear presentation in our paper, and the importance of research on test-time learning. Below, we address their concerns and questions:
>
> ### **W1: The reviewer is interested in more details about the test-time learning of long-context at inference time**
> We first want to clarify that Equation (3) describes only the training objective of PERK, not the test-time learning objective at inference time. When we train a PERK model through optimization-based meta-learning, we are teaching the model how to do test-time learning. Thus, we have to have labeled reasoning tasks r = (K,q,y) and use the answer y solely in the outer loop training to learn an initialization of the adapter and inner-loop hyperparameters that make self-supervised adaptation useful for reasoning.
>
> At test time, however, we do not have an outer optimization loop, and thus we do not optimize Equation (3), nor do we require labels. We only run the inner-loop self-supervised adaptation to encode the long context into the adapter (“memory scratchpad”). After this step, we answer any number of questions q about K using a forward pass of the adapted model, without further gradient updates or access to question labels. Thus, the test-time learning part is purely self-supervised on the long context, and the labels in Equation (3) are used only offline during full training to ensure that this self-supervised encoding is optimized for downstream reasoning.
>
> To summarize, **PERK does not require question labels for test-time learning at inference.** The complete inference algorithm is as follows:
> 1. **Input Long Context**: The model receives a long context (split into a batch of sequences).
> 2. **Test-Time Learning (Encoding Long Context)**: The model performs **self-supervised test-time learning** on the long context by minimizing the **next-token-prediction** objective (i.e., token-level NLL loss). The optimization internalizes the long context into the memory scratchpad (LoRA adapter). Note that the model **only sees the long context without access to the question or the answer at this step**.
> 3. **Input Question**: The model receives a question from the user.
> 4. **Reasoning**: The model performs a **forward pass** through the **backbone LLM** and the **memory scratchpad** to predict the answer to the input question.
>
> Note that the gold answers (labels) to the question **never appear in the inference pipeline**. They are **only used for evaluation** when we compute the **EM and F1 metrics** of the model predictions.

---

> > ### Author Response · Authors · 2025-11-24
> >
> > ### **W2: The reviewer is interested in the justification for the Length Generalization in Figure 4**
> > In PERK, the model never computes attention over the full 32k context window, so the fact that the Qwen base model was pretrained over sequences of this length is peripheral. The global context (e.g., 32K tokens) is segmented into smaller chunks of fixed length (256 tokens in our experiments). At training, when the sequences are all 8K tokens long, each segment (of length 256) is processed independently as a part of a batch of size 32 during the inner loop encoding phase. At inference time, to model a sequence of size 32K, the batch size is simply increased to 128 segments (of length 256).
> >
> > Rather than relying on attention as the mechanism to encode long-context information, PERK compresses information into the LoRA adapter through gradient-based updates. This means PERK's **effective context window** (the longest sequence over which attention is computed, in this case 256) remains constant regardless of the global context length (e.g., 32k).
> >
> > However, for FT-ICR, the reviewer is right that this is relevant, as the model does see 32k context lengths. We will reframe “length generalization” as "length robustness", highlighting this study instead as Qwen's robustness across different context lengths within its technically claimed context limit. This framing more precisely highlights PERK's advantage: rather than depending on the global context window, PERK learns a self-supervised compression strategy that generalizes to unseen context lengths at test time as highlighted by its extension to 64k/128k contexts in Table 1, that far exceeds the performance of FT-ICR baselines, even the ones trained specifically for longer contexts (Qwen2.5-7B-Instruct-1M, ProLong-8B-Instruct-512K).

---

> > > ### Author Response · Authors · 2025-11-24
> > >
> > > ### **W3: The reviewer is interested in the training cost of PERK vs. FT-ICR**
> > > Both PERK and FT-ICR can be trained using a single H100 GPU with 93GB of VRAM. With the longest training length of 8K, FT-ICR’s training speed is 2 training iterations per second, and PERK’s speed is 0.7 training iterations per second. However, we emphasize that the training cost here is amortized as PERK doesn’t need the outer loop training at inference time and displays comparable inference runtime to FT-ICR (Figure 7b originally, Figure 6b revised manuscript). However, as we note in Figure 4, the performance of FT-ICR finetuned on 8K contexts is much worse than PERK trained on 8K contexts. We will add a plot comparing training cost with performance for different context lengths across both methods.

---

> > > > ### Comment · Reviewer_sFKq · 2025-11-24
> > > >
> > > > Thank you for the response. I still have concerns about the claim of length extrapolation in Figure 4.
> > > > * Definition of Length Extrapolation: the inference length is longer than the training length
> > > > * The Qwen pretrain length is 32K. Therefore, there is no length extrapolation.
> > > > * To address the concerns: validate the Figure  4 model on a length of more than 32K.

---

> > > > > ### Author Response · Authors · 2025-11-24
> > > > >
> > > > > We thank the reviewer for their quick response.
> > > > >
> > > > > For the performance of FT-ICR on a length of more than 32K, we would like to point the reviewer to Table 1 in our paper, where we show length generalization of PERK on 64K and 128K, compared to FT-ICR. Our results show that PERK extrapolates substantially better than FT-ICR at 64K and 128K. Even after we apply SOTA training-free length generalization methods, Yarn and DCA, to the FT-ICR model, the baseline performance is still far below PERK for both QA1 and QA2 (around 35% lower on average).
> > > > >
> > > > > As we mentioned in our response above, we will revise the paper and reframe “length generalization” as "length robustness" (removing mentions to ”extrapolation”) to highlight the analysis of Qwen's robustness across different context lengths within the 32K context window.

---

> > > > > > ### Comment · Reviewer_sFKq · 2025-11-25
> > > > > >
> > > > > > Thank you for the response. The score is updated as the concern is addressed.

---

> > > > > > > ### Author Response · Authors · 2025-11-25
> > > > > > >
> > > > > > > We thank the reviewer for their encouraging response and constructive suggestions! We are grateful to the reviewer for raising their score from 4 to 6.

---

### Official Review · Reviewer_d52o · 2025-10-31

**Soundness:** 3
**Presentation:** 4
**Contribution:** 3
**Rating:** 6
**Confidence:** 4

**Summary:**

This paper introduces PERK (Parameter-Efficient Reasoning over Knowledge), a novel meta-learning framework that enhances large language models’ (LLMs) ability to perform long-context reasoning. Instead of relying solely on in-context learning or expensive fine-tuning, PERK allows models to encode long contexts through gradient updates at test time, using lightweight Low-Rank Adapters (LoRA) as memory modules. The approach employs two nested optimization loops: the inner loop encodes context into LoRA parameters, while the outer loop learns how to use these adapted parameters for reasoning. Across multiple benchmarks, including Needle-in-a-Haystack, HotpotQA, and a new Drops-in-the-Ocean dataset, PERK consistently outperforms long-context fine-tuning baselines and specialized long-context models, showing strong robustness to reasoning complexity, length extrapolation, and positional variation in relevant information.

**Strengths:**

- The paper offers a wide-ranging experimental study across multiple long-context reasoning benchmarks, including Needle-in-a-Haystack, HotpotQA, TriviaQA, and the newly introduced Drops-in-the-Ocean (DIO) dataset. The introduction of DIO, which features distributionally similar distractors to better test reasoning precision, strengthens the evaluation by addressing limitations of prior benchmarks. Comparisons against both open-source and commercial LLMs convincingly demonstrate PERK’s superiority.

- PERK exhibits remarkable robustness to reasoning complexity, unseen context lengths (up to 128K tokens), and positional shifts of relevant information. Its consistent performance across model families (GPT, Qwen, LLaMA) and scales indicates that the method learns transferable reasoning strategies rather than overfitting to a particular model or data distribution.

- The approach is built on solid meta-learning foundations, combining bi-level optimization with truncated gradient unrolling to balance effectiveness and efficiency. Its LoRA-based parameter-efficient adaptation enables scalable test-time learning without full model updates, making PERK both methodologically principled and computationally practical.

**Weaknesses:**

- The authors don't mention any public release of the code

- In section 2 of the appendix I see you only ran you experiments setting temperature to 0. Have you tried other values and what results have you found? I'd be curious to see the robustness of PERK at different decoding parameters

- While I didn't find any major weakness, what prevented me from giving a strong accept to this paper is a clear metric of inference cost for PERK. For this method to be useful to researchers and ML practitioners, data on inference cost is fundamental. While a direct comparison with fine-tuning methods is not fair, at least some stats on time needed by PERK would make the paper contribution much more comprehensive

Overall, I think PERK makes a solid contribution and I vote for its inclusion at the conference as is. With additional inference cost analysis, I'd be happy to further increase my score

**Questions:**

- Will the code be publicly released?

- Reading PERK made me thing about PATHGOOSE [1], where at each LLM token the authors propose to route to a LoRA module that has been finetuned for a specific task. I haven't seen a comprehensive description of any future work you envision for PERK and was wondering if some variant of PATHGOOSE is something you thought about to improve efficiency.


[1] Mohammed Muqeeth, Haokun Liu, Yufan Liu, Colin Raffel, Learning to Route Among Specialized Experts for Zero-Shot Generalization,

---

> ### Author Response · Authors · 2025-11-24
> **Response to Reviewer d52o**
>
> We thank the reviewer for recognizing our experiments as comprehensive, the novel benchmark contribution with the Drops-in-the-Ocean (DIO) dataset, our method as principled and computationally practical, and the strong generalization and robustness of our method. Below, we address their concerns and questions:
>
> ### **W1 & Q1: The reviewer is interested in the public release of the code**
> We thank the reviewer for their interest in our work. Our code will be publicly released after the double-blind submission phase completes.
>
> ### **W2: The reviewer is interested in PERK’s performance under different temperatures**
> We conducted PERK under different temperatures (1.0, 0.7, 0.5, 0.3, 0.0) and top-p values (0.9, 0.7, 0.5, 0.3, 0.0) on BabiLong’s QA2 (two-hop-reasoning) task with an 8K context length. We find the performance of PERK stays consistent with a small variance: ±0.953% for temperature and ±1.263% for top-p.

---

> > ### Author Response · Authors · 2025-11-24
> >
> > ### **W3: The reviewer is interested in the inference time cost of PERK**
> > We would like to point the reviewer to our Appendix C (original paper) and Section 5.3 (revised manuscript) for a detailed analysis of inference memory and time complexity. Our analysis shows that PERK scales more efficiently in memory and runtime on extremely long contexts. While FT-ICR is initially more efficient, its memory and runtime grow rapidly. PERK can manage growing contexts through gradient accumulation, which, while increasing runtime, reduces the memory footprint. By balancing memory and runtime through tuning the gradient accumulation steps, PERK costs less memory and runs faster than FT-ICR once the context length is longer than 64K tokens. We demonstrate that PERK provides a practical path to handle extreme context lengths efficiently in both memory and runtime.

---

> > > ### Author Response · Authors · 2025-11-24
> > >
> > > ### **Q2: The reviewer asks about potential future steps for improving PERK’s inference efficiency**
> > > There are interesting parallels to PATHGOOSE, and we have been thinking about related approaches to improve PERK's efficiency. We are exploring the use of specialized CUDA kernels similar to S-LoRA [1] and Punica [2] to enable efficient batched inference with PERK. These frameworks allow a batch of requests to carry their own unique LoRA ranks and weights while sharing the base model. These kernels would allow high-throughput serving where one can batch a query regarding Context A (with Adapter A) together with a query regarding Context B (with Adapter B) in the same forward pass. We can leverage vLLM's integration of Punica/S-LoRA logic for straightforward implementation. This approach is conceptually similar to PATHGOOSE's routing mechanism but adapted for PERK's dynamically-generated, context-specific adapters rather than static task-specific modules.
> > >
> > > #### **Reference**
> > > [1] Sheng, Y., Cao, S., Li, D., Hooper, C., Lee, N., Yang, S., ... & Stoica, I. (2023). S-lora: Serving thousands of concurrent lora adapters. arXiv preprint arXiv:2311.03285.
> > >
> > > [2] Chen, L., Ye, Z., Wu, Y., Zhuo, D., Ceze, L., & Krishnamurthy, A. (2024). Punica: Multi-tenant lora serving. Proceedings of Machine Learning and Systems, 6, 1-13.

---

> > > > ### Comment · Reviewer_d52o · 2025-11-25
> > > >
> > > > I thank the authors for their work on the rebuttal. I think the inference-time cost analysis is a great addition to the original contribution and it shows the great promise of the PERK. I decide to increase my score to 8

---

> > > > > ### Author Response · Authors · 2025-11-25
> > > > >
> > > > > We appreciate the reviewer's helpful suggestions and encouraging response to our rebuttal! We are grateful to the reviewer for raising their score from 6 to 8.

---

### Official Review · Reviewer_bCFY · 2025-10-31

**Soundness:** 3
**Presentation:** 3
**Contribution:** 3
**Rating:** 6
**Confidence:** 3

**Summary:**

This paper introduces PERK, a method to improve reasoning over long contexts.

Perk learns from the context at test time by updating a small, efficient memory module (using LoRA) while keeping the main model frozen. It uses a two-step training process: first, rapidly encodes the context, and then optimizes how the model uses that info to answer questions.

Experiments on long context benchmarks show PERK significantly outperforms standard long-context finetuning.

**Strengths:**

- Paper is well written, and the results are nice
- PERK demonstrates strong generalization to long context extrapolation (e.g., training on 8K, testing on 128K) and superior robustness to positional biases in the relevant information.
- The idea of using a LoRA adapter as a differentiable memory module for context is a good alternative to ICL
- The use of LoRA and TGU is nice, and seems well executed.

**Weaknesses:**

- The biggest concern is the added complexity. The gains are nice, but I'm not sure they justify using this method over FT-ICR.
- Related to the added complexity, in general the inference time is significantly longer than ICR (at least up to 34k according to fig 7)
- PERK is designed for length generalization, as noted in the experiments (train on 8k, eval on 32k). However, FT-ICR is not suited for this kind of evaluation, as is well known. So I'm not sure this is the best baseline to compare against. Especially since we can fine-tune on 32K lengths, I think it would be nice to see an FT-ICR training with 32k length.
- Permutation-invariance can be detrimental for certain tasks (though maybe rare).

**Questions:**

- One of the mentioned strengths is that it encodes context as permutation-invariant representations, but that might not always be a strength. What if there is a query about positional information? The benchmarks do not consider that.
- In figure 6, you show that you encode context + question before the LoRA updates. This means that for each new question, you need to do the update procedure. Is there a way to encode only context so that you can ask multiple questions?

---

> ### Author Response · Authors · 2025-11-24
> **Response to Reviewer bCFY**
>
> We thank the reviewer for recognizing our paper as well-written with nice results, our method as well executed, our contributions on strong length generalization and positional robustness, and our idea of using test-time learning with LoRA as a good alternative to ICL. Below, we address their concerns and questions:
>
> ### **W1 & W2: The reviewer is interested in the trade-off between performance and complexity**
> We appreciate this feedback and the opportunity to clarify the value proposition of PERK. First, the empirical gains demonstrated in our work are substantial and address critical limitations of existing approaches. Specifically, PERK achieves up to 20% absolute improvement over FT-ICR across multiple benchmarks, and importantly, these gains become even more pronounced in challenging scenarios:
>
> 1. **Length Generalization**: PERK trained on only 8K-token contexts successfully extrapolates to 64K and 128K tokens, outperforming specialized long-context models trained on 256K–512K tokens. This represents a significant practical advantage, as training on shorter contexts is substantially more resource-efficient.
> 2. **Positional Robustness**: FT-ICR exhibits performance drops of up to 90% when relevant information shifts position at test time, whereas PERK maintains consistent performance across all positional distributions, a crucial property for real-world applications where information location cannot be controlled.
> 3. **Competitive with Commercial Models**: PERK matches or exceeds GPT-4.1 and Gemini-1.5-pro on several reasoning tasks despite using significantly smaller base models.
>
> Regarding complexity, we acknowledge that the current implementation introduces additional training overhead compared to FT-ICR. However, we note that the development and optimization of today's efficient LLM training and inference required a sustained effort across many research and engineering projects. We are confident that, given the demonstrated performance benefits of test-time learning for long-context reasoning, similar efforts will build upon PERK to improve both training and inference efficiency.
>
> In that spirit, we have already begun exploring approaches to reduce complexity. For example, using activation checkpointing and forward gradients to reduce memory overhead during training, while implementing data and context parallelism to improve inference efficiency. Our scalability analysis (Appendix C original paper, Section 5.3 revised manuscript) also shows that PERK already provides more efficient memory and runtime scaling than FT-ICR at extreme context lengths (64K–128K tokens), suggesting a favorable trajectory for practical deployment.
>
> Long-context reasoning is becoming increasingly critical for applications such as agentic LLMs and deep research systems. We believe PERK offers a promising and unique direction for building effective long-context reasoning agents by enabling models to parametrically encode context at inference time.

---

> > ### Author Response · Authors · 2025-11-24
> >
> > ### **W3: The reviewer is interested in FT-ICR’s performance when finetuned with 32K length**
> > Following the reviewer’s recommendation, we conduct additional experiments training FT-ICR (Qwen-2.5-0.5B) on 32K-token contexts and evaluating at the same length. On BabiLong QA2 (two-hop reasoning), we find that FT-ICR, finetuned with a 32K length, still significantly underperforms PERK. Qwen FT-ICR (32K) got only 49.4%, while PERK got 80.9%. The results show that although finetuning on 32K length sequences directly improves FT-ICR from 39.4% to 49.4%, PERK’s extrapolation still outperforms the in-distribution finetuning of FT-ICR, with a large performance gap of 31.5%.

---

> > > ### Author Response · Authors · 2025-11-24
> > >
> > > ### **W4 & Q1: The reviewer is curious about PERK’s performance on tasks that are not permutation-invariant**
> > > We agree that naive permutation-invariance could be problematic for tasks requiring positional or temporal information. However, PERK addresses this through explicit indexing in the prompt formatting, which allows the model to learn implicit ordering during the encoding phase. Specifically, when chunking a long context, we can preserve ordering information by indexing each chunk. For example, if we split a long document into segments A, B, C, D, E, we format them as: ["Document 1: A", "Document 2: B", …, "Document 5: E"]. During the inner loop adaptation with the NLL objective, the LoRA parameters learn to associate these indices with their content, effectively encoding the ordering information.
> > >
> > > In our paper, this capability is directly evaluated by the BabiLong QA3 task, which explicitly requires episodic memorization and reasoning over ordered events. Our experimental results show that PERK with this indexing scheme achieves strong performance on QA3 (60.4% on 8K contexts, 38.7% on 32K contexts extrapolation, Table 5), substantially outperforming FT-ICR (37.5% and 12.1% respectively). This validates that the explicit indexing successfully enables PERK to handle tasks requiring positional and temporal information.
> > >
> > > We will clarify this mechanism more explicitly in the revised manuscript to avoid confusion about PERK's handling of ordered information.

---

> > > > ### Author Response · Authors · 2025-11-24
> > > >
> > > > ### **Q2: The reviewer wonders whether PERK could encode a context once and answer multiple questions**
> > > > This question highlights an important capability of PERK. PERK already supports encoding the context once and then answering multiple questions about that encoded context. The inner loop adaptation (encoding stage) operates solely on the context K, updating the LoRA adapter parameters to compress the contextual information. Once this encoding is complete, the model can answer any number of questions about that context using the adapted parameters, without requiring re-encoding.
> > > >
> > > > In fact, our Student Records dataset is specifically structured this way: a single database (the context) is encoded once, and then multiple different questions, spanning Recall, Relation, and Aggregation tasks, are asked about that same encoded context. This demonstrates PERK's ability to amortize the encoding cost across multiple queries.
> > > >
> > > > We acknowledge that the original Figure 6 may have caused confusion by visually bundling the question together with the context in the encoding stage. To address this, we have revised the figure (Figure 7, revised manuscript) to clearly separate the two stages: (1) Encoding stage (test-time learning): Only the context is compressed via updating the Memory Scratchpad. (2) Reasoning stage: Questions are presented to the model after encoding is complete. Our revised figure makes explicit that the model only sees questions after the context has been fully encoded into the adapter parameters.

---

### Author Response · Authors · 2025-12-02
**General Response**

## **Paper Summary**
We propose PERK, a scalable approach for encoding long contexts via gradient updates at test time. PERK operates through two nested optimization loops: the inner loop compresses contexts into a LoRA adapter using a self-supervised NLL objective, while the outer loop learns to leverage this compressed representation for downstream reasoning. Our evaluations on real-world and synthetic long-context reasoning tasks show that PERK significantly outperforms standard long-context finetuning, achieving absolute performance gains of up to 20%. PERK generalizes across model scales and families, matching or surpassing specialized long-context LLMs. Finally, PERK demonstrates strong robustness to reasoning complexity, length extrapolation (16× generalization from 8K to 128K), and varying positions of relevant information within contexts.

## **Rebuttal Summary**

We thank the reviewer for their constructive feedback and suggestions. We appreciate their recognition of our work’s strengths in:

1.  **Novelty & Impact:** Novel framing of long-context reasoning as test-time learning (MTjy), Test-time learning with LoRA as a good alternative to ICL (bCFY), Important research direction in test-time learning (sFKq)
2.  **Method Quality:** Method is well executed (bCFY), Principled and computationally practical approach (d52o), scalability (MTjy), model-agnosticism (MTjy)
3.  **Performance & Robustness:** Strong length generalization (bCFY, d52o, sFKq), Positional robustness (bCFY, d52o), Empirical improvements and robustness (MTjy), Strong generalization and robustness (d52o)
4.  **Evaluation & Experiments:** Comprehensive experiments (d52o), Strong evaluation methodology (MTjy, Novel benchmark contribution with Drops-in-the-Ocean (DIO) dataset (d52o)
5.  **Presentation:** Well-written paper with nice results (bCFY), Clear presentation (sFKq)

Major reviewer concerns focused on the inference cost of PERK vs. finetuned in-context reasoning (FT-ICR) and the complexity-performance trade-off. Reviewers also requested additional baselines for length generalization: (1) evaluating FT-ICR on contexts longer than 32K, and (2) finetuning on 32K-token sequences directly. We clarified that (1) was already reported in Table 1, and conducted (2) during the discussion period, PERK still outperforms by 31.5%. Reviewers also requested ablations on decoding parameters, learnable learning rates, and weighted NLL loss; we provided all additional studies during the discussion period.

**Rebuttal Outcome:** During the initial discussion period, we addressed these concerns, resulting in two score increases (**d52o: 6 to 8, sFKq: 4 to 6**). Scores we had during the period were: **[6, 8, 6, 8]**.

Below, we summarize reviewer feedback and our responses for each reviewer in more detail.

---

> ### Author Response · Authors · 2025-12-02
> **Reviewer bCFY Summary**
>
> ### **W1 & W2: The reviewer asked about the trade-off between performance and complexity**
> Our scalability analysis (Appendix C, original; Section 5.3, revised) shows PERK achieves a runtime comparable to or more efficient than in-context baselines (i.e., FT-ICR) starting at 4K context length, and eventually provides more efficient memory and runtime scaling than FT-ICR at extreme context lengths (64K–128K tokens). PERK achieves up to 20% absolute improvement over FT-ICR, with critical advantages in length generalization and positional robustness, justifying our contribution.
>
> [https://openreview.net/forum?id=qxDTe8fIyA&noteId=kWJmj54rJI]
>
> ### **W3: The reviewer was interested in FT-ICR’s performance when finetuned with 32K length**
> We conducted the requested experiment. FT-ICR trained on 32K still significantly underperforms PERK on BabiLong QA2 with a 31.5% gap. This confirms PERK's extrapolation outperforms even in-distribution FT-ICR training.
>
> [https://openreview.net/forum?id=qxDTe8fIyA&noteId=gyfen2JCwo]
>
> ### **W4 & Q1: The reviewer was curious about PERK’s performance on tasks that are not permutation-invariant**
> PERK preserves ordering via explicit chunk indexing (e.g., "Document 1: A", "Document 2: B"). This is validated on BabiLong QA3 (which requires understanding timelines across documents), where PERK achieves 60.4% vs. FT-ICR's 37.5% at 8K. We clarified this mechanism in the revised manuscript.
>
> [https://openreview.net/forum?id=qxDTe8fIyA&noteId=5GdjU1WngQ]
>
> ### **Q2: The reviewer wondered whether PERK could encode a context once and answer multiple questions**
> PERK already supports encoding the context once and answering multiple questions. The Student Records dataset represents this task format. We revised Figure 6 to clarify that the encoding process does not bind to a particular question; rather, any questions about the context can be asked once the encoding completes.
>
> [https://openreview.net/forum?id=qxDTe8fIyA&noteId=mgFNYE41XD]
>
> **Outcome:** The response period ended before the interaction could be resolved.

---

> > ### Author Response · Authors · 2025-12-02
> > **Reviewer d52o Summary**
> >
> > ### **W1 & Q1: The reviewer asked whether the code would be publicly released.**
> > We will link the publicly released code after the double-blind submission phase completes.
> >
> > [https://openreview.net/forum?id=qxDTe8fIyA&noteId=BgRwlIi5lC]
> >
> > ### **W2: The reviewer was interested in PERK’s performance under different temperatures**
> > We tested PERK across temperatures (0.0–1.0) and top-p values (0.0–0.9) on BabiLong QA2 at 8K context. Performance remains highly consistent: ±0.953% variance for temperature, ±1.263% for top-p.
> >
> > [https://openreview.net/forum?id=qxDTe8fIyA&noteId=BgRwlIi5lC]
> >
> > ### **W3: The reviewer asked about the inference time cost of PERK**
> > We provide detailed memory and runtime analysis in Appendix C (original) / Section 5.3 (revised). PERK scales more efficiently at extreme lengths with lower memory and faster runtime than FT-ICR beyond 64K tokens via balancing gradient accumulation.
> >
> > [https://openreview.net/forum?id=qxDTe8fIyA&noteId=OJSwbuxq4r]
> >
> > Outcome: **Reviewer d52o** thinks that the inference-time cost analysis is a great addition to the original contribution, and it shows the **great promise of PERK**. Initially, **they raised their score from 6 to 8**.
> >
> > [https://openreview.net/forum?id=qxDTe8fIyA&noteId=5dfLmN8Hnm]

---

> > > ### Author Response · Authors · 2025-12-02
> > > **Reviewer sFKq Summary**
> > >
> > > ### **W1: The reviewer asked whether answer labels are used for the test-time learning**
> > > At test time, PERK runs **only the inner loop adaptation**, which is purely self-supervised NLL on the long context, so no answer labels are used. The inference pipeline is: (1) encode context via self-supervised adaptation, (2) answer questions using encoded context with forward passes, so **no answer labels are required in the whole process**. Answer labels are only used during training.
> > >
> > > [https://openreview.net/forum?id=qxDTe8fIyA&noteId=9U5l96EZPY]
> > >
> > > ### **W2: The reviewer was interested in the justification for the Length Generalization in Figure 4**
> > > The reviewer requested us to validate length generalization by evaluating on contexts that are longer than 32K tokens. **Our original draft already reports PERK and multiple baselines’ generalization performance beyond 32K length in Table 1**. Our results show that **PERK extrapolates substantially better than FT-ICR at 64K and 128K**. We reframed “length generalization” as "length robustness" (removing mentions of ”extrapolation”) for analysis within the 32K Qwen context window in the revised version.
> > >
> > > [https://openreview.net/forum?id=qxDTe8fIyA&noteId=I33v58XqV3]
> > > [https://openreview.net/forum?id=qxDTe8fIyA&noteId=FypauAVgbB]
> > >
> > > **Outcome**: **Reviewer sFKq** acknowledge that the concern is addressed. **They initially raised their score from 4 to 6.**
> > >
> > > [https://openreview.net/forum?id=qxDTe8fIyA&noteId=idZy6EzOaw]

---

> > > > ### Author Response · Authors · 2025-12-02
> > > > **Reviewer MTjy Summary**
> > > >
> > > > ### **W1 & Q1: The reviewer was interested in the inference cost of PERK vs. FT-ICR**
> > > > While PERK does increase latency at shorter lengths, the break-even point for runtime occurs around 4K-8K tokens, where PERK's more efficient scaling begins to compensate for the gradient update overhead. PERK offers a memory-runtime trade-off via gradient accumulation tuning. At 64K+, PERK is faster and more memory-efficient. We believe that substantial performance gains, length extrapolation, and positional robustness justify the latency trade-off.
> > > >
> > > > [https://openreview.net/forum?id=qxDTe8fIyA&noteId=g98B8K1Eod]
> > > >
> > > > ### **W2 & Q4: The reviewer asked for ablation studies on PERK’s algorithmic design**
> > > > Ablations provided in Appendix D (original) / Appendix C (revised) for LoRA rank, inner loop steps, and TGU truncation. We conducted additional ablation: Learnable LR is crucial (training fails without it); weighted loss improves performance from 83.9% to 98.1% on QA2.
> > > >
> > > > [https://openreview.net/forum?id=qxDTe8fIyA&noteId=O91eTWq4qT]
> > > >
> > > > **Outcome**: The response period ended before the interaction could be resolved.

---

> > > > > ### Author Response · Authors · 2025-12-02
> > > > > **Thank You ACs for your Time and Consideration**
> > > > >
> > > > > **We sincerely thank the Area Chairs for their time and effort in evaluating our submission. We appreciate the opportunity to address reviewer concerns and improve our manuscript through this process.**

---

### Meta-Review · Area_Chair_WFE2 · 2026-01-05

**Summary:**

The authors present a new method for compressing long horizon reasoning. The approach seems competitive with models which are explicitly trained on long context. Performance appears to be competitive with long-horizon commercial models.

**Reviewer Concerns:**

Reviewers were concerned with inference speed and code release. Code release was promised, and detailed inference speed evaluations were provided.

**Reviewer Scores:**

Given the author's thorough answer to all the reviewer concerns, I believe that most scores would have been raised to an 8.

---

### Decision · Program_Chairs · 2026-01-26

Accept (Poster)